**EMBO** *reports*

# The invasion pore induced by *Toxoplasma gondii*

Yuto Kegawa [1,3], Frances Male[2,3], Irene Jiménez-Munguía [1], Paul S Blank [1], Elena Mekhedov[1], Gary E Ward [2✉] & Joshua Zimmerberg [1✉]

## Abstract

The parasite *Toxoplasma gondii* invades its host cell only after secreting proteins such as invasion-requisite RON2 that inserts into the host cell membrane to establish the moving junction. Electrophysiological recordings at sub-200 µs resolution show a transient increase in host cell membrane conductance following parasite exposure. Transients always precede invasion, but parasites depleted of RON2 generate transients without invading. Thus RON2 is not essential for transient generation. Time-series analysis developed here and applied to the 910,000 data point transient dataset reveal multiple quantal conductance changes in the parasite-induced transient, consistent with rapid insertion, then slower removal, blocking, or inactivation of potential pore components. Quantal steps for wild-type RH strain parasites have a principal mode with Gaussian mean of 0.26 nS, similar in step size to the pore forming protein EXP2, part of the PTEX translocon of malaria parasites. Without RON2 the quantal mean (0.19 nS) is significantly different. Because we observe no parasite invasion without poration, the term "invasion pore" is proposed to describe this transient breach in host cell membrane barrier integrity during invasion.

**Keywords** Parasite; Membrane Poration; Rhoptry Secretion; Cell Membrane; Whole-Cell Patch-Clamp
**Subject Categories** Membranes & Trafficking; Microbiology, Virology & Host Pathogen Interaction

See also: F Male et al

## Introduction

Protozoan pathogens belonging to the phylum *Apicomplexa* cause life-threatening diseases such as malaria, toxoplasmosis, and cryptosporidiosis in humans and other animals that are important global health burdens with few effective vaccines and limited drugs for treatment (Smith et al, 2021; Striepen, 2013; WHO, 2023). They are obligate intracellular parasites; the intracellular stage of their life cycles begins after host cell invasion. To invade, two types of secretory organelles unique to *Apicomplexa* are employed: micronemes and rhoptries (Bullen et al, 2019; Carruthers and Sibley, 1997; Dubremetz, 2007; Reviewed in Cova et al, 2022). These organelles contain proteins whose highly coordinated secretion is essential to successful invasion. In *Toxoplasma gondii* tachyzoites, the life cycle stage associated with acute infection, micronemes are abundant organelles mostly located around the parasite's apical pole (Carruthers and Sibley, 1999). The rhoptries are the largest secretory organelles in the tachyzoite and consist of a rhoptry neck and bulb (Counihan et al, 2013; Proellocks et al, 2010). Between the apical tip of the rhoptry and the parasite plasma membrane lies a small apical vesicle (AV). Current understanding of rhoptry secretion involves fusion of the rhoptry with the AV and fusion of the AV with the parasite plasma membrane, resulting in rhoptry exocytosis (Aquilini et al, 2021; Mageswaran et al, 2021; Sparvoli et al, 2022). Proteins secreted from a third group of secretory organelles, the dense granules, play an important role in parasite intracellular survival and pathogenesis (Gold et al, 2015; Griffith et al, 2022; Bitew et al, 2024), but there is no evidence that these secreted GRA proteins function in the early stages of invasion.

One of the most enigmatic of the molecular events during *T. gondii* invasion is the appearance of rhoptry proteins in the host cell cytoplasm during both normal and incomplete ("abortive") invasions (Håkansson et al, 2001; Ravindran and Boothroyd, 2008; Koshy et al, 2012; Lamarque et al, 2014). The presence of rhoptry proteins in the host cell cytoplasm seems to violate a fundamental cell biological law governing conservation of membrane and secretory protein topology, i.e., that protein domains co-translationally inserted into the lumen of the ER remain lumenal throughout the secretory pathway, including within secretory granules, and once released from those granules, face the extracellular space. Accordingly, the lumen of the rhoptry is topologically equivalent to the extracellular space and not the host cell cytoplasm. Proteins released from the lumen of the rhoptry can change their topology only by traversing a limiting membrane such as the host cell plasma membrane. The rhoptry neck protein RON2 presents a particularly unique challenge, in that RON2 is a transmembrane protein that is somehow inserted into the host cell plasma membrane after exocytosis, where it serves as a ligand to which AMA1 in the *Toxoplasma* plasma membrane binds and contributes to the formation of a complex of translocated proteins necessary for parasite internalization (the moving junction) (Alexander et al, 2005; Besteiro et al, 2009; Lamarque et al, 2011; Srinivasan et al, 2011; Lamarque et al, 2014). Thus, secreted RON2 is physically proximal to the site of parasite entry.

[1]Section on Integrative Biophysics; Division of Basic and Translational Biophysics, Eunice Kennedy Shriver National Institute of Child Health and Human Development (NICHD), National Institutes of Health (NIH), Bethesda, MD, USA. [2]Department of Microbiology and Molecular Genetics, University of Vermont Larner College of Medicine, Burlington, VT, USA. [3]These authors contributed equally: Yuto Kegawa, Frances Male. ✉E-mail: Gary.Ward@uvm.edu; Joshua.Zimmerberg@nih.gov

A transient increase in host cell conductance precedes parasite internalization (Suss-Toby et al, 1996). In a companion paper (Male et al, 2025), this transient increase in host cell ionic permeability (membrane conductance) is also detected as an influx of extracellular $Ca^{2+}$ and is shown to depend on rhoptry exocytosis. $Ca^{2+}$ entry into the host cell occurs at the site of parasite apical contact (Male et al, 2025), which is also the site of rhoptry protein discharge. Since RON2 is the only known component of the moving junction that is inserted as a transmembrane protein into the host cell plasma membrane, it was reasonable to hypothesize that the reason the RASP2 mutant (i.e., a mutant lacking the rhoptry apical surface protein 2 -RASP2, which is a protein regulating the rhoptry protein discharge) blocked poration (Male et al, 2025) was that RON2 was not delivered to the host cell membrane.

Here, we test this hypothesis by measuring host cell plasma membrane conductance and $Ca^{2+}$ entry during *T. gondii* invasion. The hypothesis is ruled out because RON2 was not required for host cell poration. A detailed electrophysiological analysis of the conductance transient revealed the presence of multiple stepwise changes in conductance throughout its trajectory, consistent with a multiple-pore model in the host cell membrane. Based on these results, we propose that these stepwise changes in conductance represent the incorporation or formation of discrete membrane structures we term "invasion pores" that may function as a transient breach in the host cell plasma membrane, allowing rapid rhoptry protein translocation into the host cytosol.

## Results

### Each invading wild-type *T. gondii* tachyzoite produces an individual conductance transient

Transients were collected with the *T. gondii* wild-type (WT) RH strain using high bandwidth direct current measurements of host cells (COS1) under voltage-clamp conditions in the whole-cell configuration. Simultaneously, parasite invasion was monitored by DIC microscopy (Fig. 1A). Multiple parasites were delivered to the host cells to increase the probability of capturing multiple invasions and the associated transient increases in whole-cell current. Invading tachyzoites showed a clear constriction (arrowheads, Fig. 1A) and a single large current change was detected prior to the constriction of each parasite: multiple transients were recorded when multiple parasites invaded the same host cell (Fig. 1B). Parasite invasion (i.e., visible constriction) and the current transient were correlated events; no instance of invasion was observed without a transient (16 invasions with 16 transients). However, transients were also detected without subsequent parasite invasion (9/25 WT transients), likely representing what has previously been described as abortive invasion (Koshy et al, 2012; Lamarque et al, 2014) and will be discussed later. Further analysis was performed on all transients identified using the WT strain ($n = 25$). The analysis revealed that every transient possessed characteristic waveform features—a fast rise to a peak and a slower fall to a new baseline—although the magnitude and duration of the transients varied (e.g., Figs. 1C and EV1A,B). No obvious difference in transient conductance was seen based on the order of parasite entry, i.e., the first, middle, and last transients seemed similar

(Fig. 1C), suggesting that each transient represents an independent event irrespective of the number of invasions per cell. To corroborate independence, and because characteristic waveform features were similar, we evaluated all wildtype-like conductance transient recordings, including those which were controls for our companion paper (see below and Male et al, 2025; Figs. EV1A–H and EV2A,B).

### The transient conductance increase induced by wild-type parasites occurs independently of complete moving junction formation

The depletion of rhoptry neck protein 2 (RON2) causes a severe early invasion defect in most of the parasites, wherein rhoptry proteins are seen inside the host cell, but the parasite subsequently detaches (Lamarque et al, 2014). However, a small number of the parasites can still invade through an "incomplete" moving junction, meaning a moving junction with little to no RON2, RON4, and RON5, but still containing RON8 and perhaps other normal or alternative binding partners (Lamarque et al, 2014). To investigate whether RON2 insertion (and thus "complete" moving junction formation) is associated with the appearance of transients, we used both current recording experiments and the calcium influx assay (Male et al, 2025) to independently interrogate the ability of RON2 knockdown (KD-RON2) parasites to generate the conductance and calcium transients. KD-RON2 parasites do not express detectable levels of RON2, even without the ATc treatment typically used to induce knockdown, and show a severe invasion defect (Lamarque et al, 2014). While zero out of 159 untreated KD-RON2 parasites (hereafter in this paper referred to as KD-RON2) invaded COS1 cells in our electrophysiology experiments, i.e., no visible constrictions were observed by DIC microscopy, 19.5% (13.7–26.5%; 95% CI) of the same parasites generated conductance transients (Figs. 2A and EV2A,B). Similar results were seen using the calcium transient assay (Fig. 2A). KD-RON2 parasites generating conductance and calcium transients at levels similar to WT parasites stand in stark contrast to their near absence when the parasites are depleted of proteins that regulate rhoptry exocytosis (Male et al, 2025). Together, the conductance and calcium transient results rule out the hypothesis that RON2 insertion into the host cell plasma membrane is itself the poration process.

### Detailed analysis of conductance transients induced by WT and KD-RON2 parasites

The conductance transients produced by both the WT ($n = 25$) and KD-RON2 parasites ($n = 30$) were averaged and compared (Figs. 2B and EV1A,B, EV2A,B). On average, both WT and KD-RON2 parasite conductance transients show a rapid increase and then a slower decrease in conductance during the transient. To further characterize and compare the transients, change point analysis of individual transients was performed (see Methods and Fig. 3). Individual transients are described by four characteristic features (mean ± std. dev.) for WT and KD-RON2 ($n = 25$ and $n = 30$, respectively): (1) Peak conductance, calculated as the difference in conductance between the baseline and the maximum change point mean (WT: 3.40 ± 1.12 nS and KD-RON2: 3.01 ± 1.40 nS, Fig. 4A); (2) Residual conductance, calculated as the difference in conductance between the pre- and post-transient baselines (WT: 0.54 ± 0.38 nS and KD-RON2: 0.39 ± 0.33 nS, Fig. 4B); (3) Transient duration, calculated

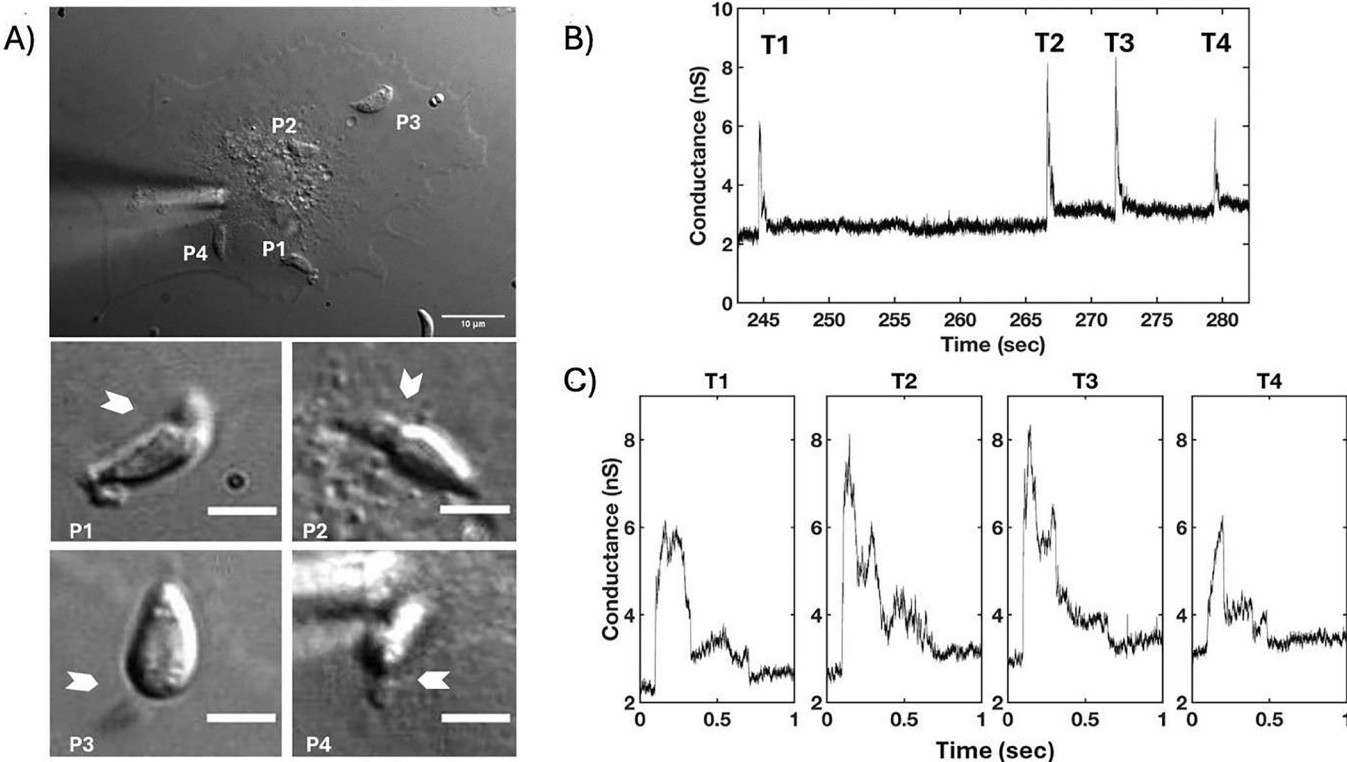

**Figure 1. Each invading *T. gondii* WT RH strain tachyzoite produces a single electrical transient.**

(**A**) DIC microscopy of *Toxoplasma* invasion of COS1 cell under visible patch pipette (upper panel; 10 µm scale bar). Four parasites (P1–P4) invaded during the recording of a whole-cell patch-clamp electrophysiology experiment. White arrowheads indicate visible constrictions observed when parasites penetrate the host cell during invasion (middle panel: P1 and P2 invasions; lower panel: P3 and P4 invasions; 3 µm scale bar panels P1–P4). (**B**) Four transients were detected from the electrophysiological recording obtained under the whole-cell configuration (−60 mV holding potential) during the invasions of the four parasites. (**C**) Expanded time record of the transients (T1–T4) shown in B) for a clearer visualization of the waveform. Source data are available online for this figure.

as the time between transient initiation and the beginning of residual conductance (WT: $322 \pm 219$ ms and KD-RON2: $374 \pm 203$ ms, Fig. 4C), and (4) Peak conductance duration, calculated as the difference in the starting and ending change point time of the maximum conductance identified (WT: $73 \pm 64$ ms and KD-RON2: $118 \pm 86$ ms, Fig. 4D). Both the WT and KD-RON2 transient waveforms begin with a rapid increase in conductance, reaching a maximum value with exponentially distributed time constants of $50 \pm 10$ ms and $51 \pm 9$ ms (mean ± std. err.; $n = 25$ and $n = 30$), followed by exponentially distributed decreases (fall times) to the residual conductance with time constants of $197 \pm 39$ ms and $203 \pm 36$ ms (mean ± std. err.; $n = 25$ and $n = 30$). For any one parasite strain (WT or KD-RON2), the rise time and fall time distributions are significantly different from each other; two-sample Kolmogorov–Smirnov test, $p_{WT} = 3.97\text{E-}04$ and $p_{KD\text{-}RON2} = 8.08\text{E-}4$; $n = 25$ and $n = 30$, respectively. Comparing the two strains, there is no evidence for differences between WT and KD-RON2 peak conductance, residual mean conductance, or mean transient duration. However, between WT and KD-RON2 strains the peak conductance duration is different (Fig. 4D). The change point analysis supports the hypothesis that the kinetic process for creating and removing the conductance pathway for WT and KD-RON2 are similar (rise and fall time distributions, peak conductance, and transient duration:

Fig. 4A–C) but the lifetime (peak duration: Fig. 4D) of the maximal conductance state is significantly longer in the absence of RON2.

## Step-like transients induced by WT and KD-RON2 parasites differ in quantal conductance

The above parameterization of the transients was consistent with similar pores produced by WT and KD-RON2 parasites, but the longer peak conductance duration observed in the averaged transients and confirmed in the analysis of the individual peak durations (Figs. 2B and 4D) led to further investigation of the abrupt changes in conductance levels in both the rising (Figs. EV1B and EV2B) and falling phases (Figs. EV1A and EV2A) of the transients. To analyze the magnitudes and numbers of the rapid, step-like changes in conductance, each transient was processed using piecewise constant (PWC) signal denoising (Methods and Figs. 3A and 5A,B). From the resulting conductance values, the density distribution of the differences in adjacent conductance levels was obtained (Figs. 3B and 5C). WT parasite-induced transients display a probability density peak at 0.26 nS, which can be considered a characteristic quantal step size (Fig. 5A,C,D). This hypothesis was further tested by both parametric model fitting and Gaussian mixture modeling, each identifying a primary Gaussian

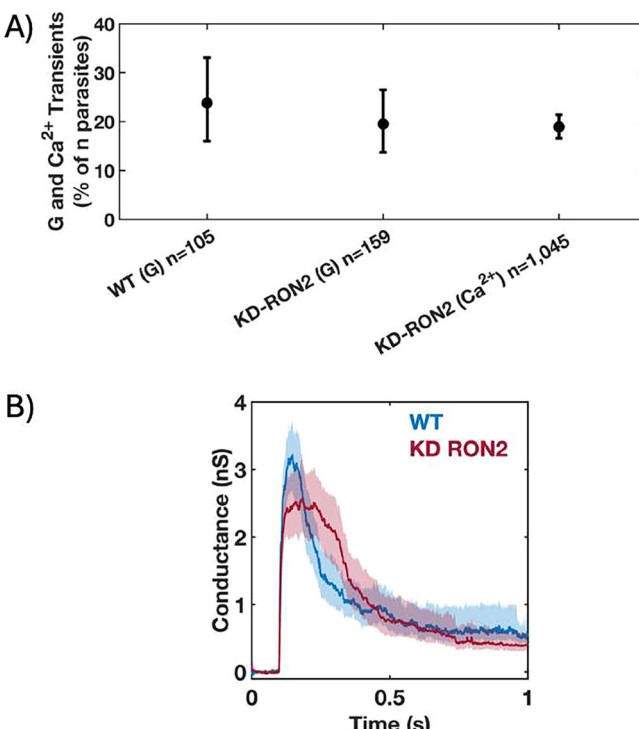

**Figure 2.  *T. gondii* invasion pore formation does not require RON2, and thus does not require complete moving junction formation.**

(A) Phenotype of untreated WT and KD-RON2 parasites evaluated using electrophysiology (G) or calcium assay ($Ca^{2+}$) data sets. The data show the percentage of the parasites presented to the host cell that generated conductance or calcium transients. Errors are Wilson, two-tailed upper and lower 95% confidence intervals. The percentage of transients observed in KD-RON2 (G) and KD-RON2 ($Ca^{2+}$) are not significantly different from WT (G) (two-proportion z-test, $z = 0.8389$, 1.227, $p = 0.40$, 0.22, respectively).
(B) Conductance waveforms for WT (blue) and KD-RON2 strains (red) (mean, solid lines; 95% CI, shadings; $n = 25$, 30, respectively). Source data are available online for this figure.

peak at 0.26 nS with a width of 0.03 nS. The same analysis was applied to the KD-RON2 conductance transients, yielding a smaller primary Gaussian peak at 0.19 nS with a width of 0.03 nS (Fig. 5B–D). The number of quantal units contributing to the averaged maximum conductance for both parasite lines are approximated by Poisson distributions with parameters 13.08 and 15.90 for WT and KD-RON2, respectively, an insignificant difference (no evidence using bootstrap differences in the Poisson parameter, alpha = 0.05); Fig. EV3A,B). Thus, knocking down RON2 does not eliminate the transient or the number of steps, but does change the step size significantly (27% smaller).

To control for the possibility that the quantal step size parameter is sample size dependent, transient datasets generated by WT and other control parasites (WT parasites in low $Ca^{2+}$ conditions and RASP2 conditional knockdown (cKD) parasites (Suarez et al, 2019) without anhydrotetracycline (ATc) treatment), each with different sample sizes, were compared. Ten additional transients were recorded in low $Ca^{2+}$ (Fig. EV1C,D), and 26 transients using the cKD-TgRASP2 -ATc parasites (Fig. EV2C,D). Both datasets showed the same characteristic waveform features as

WT. PWC analysis was performed on each individual transient recorded, and the density of conductance level changes were calculated. The primary peak quantal step values were 0.24 (0.03) nS and 0.25 nS (0.03) for WT low $Ca^{2+}$ environment and cKD-TgRASP2 -ATc, respectively (mean (width)); the means are within the uncertainty of WT (0.26 (0.03)), consistent with conductance levels acquired from parasites having similar properties and similar quantal values. For completeness, all four groups were compared using both ANOVA (parametric) and Kruskal–Wallis (non-parametric) analyses. WT, WT low $Ca^{2+}$, and cKD-TgRASP2 -ATc are each significantly different from KD-RON2 (Tukey's $p < 10^{-15}$, and Dunn's post-hoc analyses, $Q > 21$).

## Discussion

The distinctive transient increase in host cell membrane conductance detected after parasite attachment but preceding the morphological changes of invasion implies that for a brief time, ions can freely move across the host cell membrane. Ion movement (flux) is likely due to the appearance in the host cell membrane of an aqueous pathway. Pathways for passive ion flux across membranes can be classified as either protein-lined channels (termed "protein pores") or localized ruptures of the hydrocarbon continuity of the lipid bilayer (termed "lipidic pores"). The invasion transient proceeds along a series of intermediates in conductance: once initiated, the conductance rises in tens of milliseconds, reaching a peak before diminishing more slowly to a residual conductance that is approximately 10% of that peak (by one second after the initiation of the conductance transient). RON2 was not required for the transient increase of either membrane conductance or $Ca^{2+}$ entry. Consistent with an aqueous pathway for ion flux during the conductance transient is the increasing $Ca^{2+}$ flux into the host cell cytoplasm with increasing extracellular $Ca^{2+}$ concentration, as shown in our companion paper that also revealed that transients require rhoptry exocytosis (Male et al, 2025). However, extracellular calcium is not itself likely to be the major charge-carrying ion or required for either pore opening or closing, since conductance transients of similar shape and magnitude were observed in an environment with twenty times lower extracellular calcium concentration (Fig. EV4). Based on these combined data, we propose that the conductance and calcium transients observed are manifestations of a rhoptry exocytosis-dependent host cell membrane poration process (i.e., the insertion of "invasion pores"), whose purpose is to provide the pathway for rhoptry protein translocation into the host cell cytosol. This model (Fig. 6) is consistent with previous data showing that parasites lacking RON2 can deliver secreted rhoptry proteins into the host cytosol without complete moving junction formation (Lamarque et al, 2014). What follows are deductions about this poration process based on analysis of the electrophysiological data presented.

### Kinetics of the conductance transient do not match one single large channel opening and closing

Every time invasion occurs, a transient conductance increase precedes the first invasion-associated morphological changes in the parasite. However, the role that the transient plays in invasion is unknown. Here we consider molecular structures that might

      

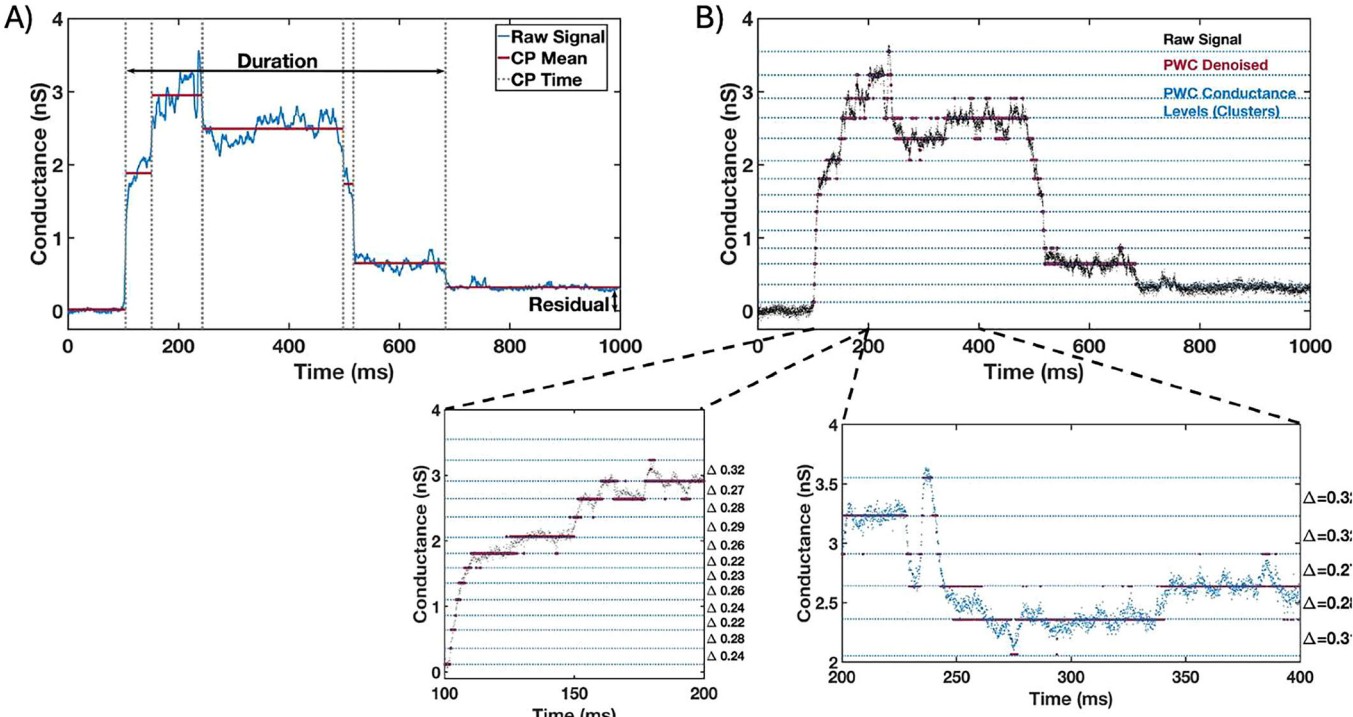

**Figure 3. Structure and analysis of a conductance transient induced by _T. gondii_ prior to invasion.**

(A) Representative change point analysis (CPA) of a transient waveform induced by a WT parasite. Time-dependent changes in conductance are represented by the gray dotted lines (CP Time) over a selected duration of 1 s. Characteristic features of the transient including the change point means (CP Mean, red), Duration, and Residual conductance (black arrows) are indicated. (B) Representative PWC analysis of the same transient in (A) showing the identified conductance levels (clusters) in blue. Expanded portions of the rising phase (100 ms) and a larger section of the transient (200 ms) are shown in the expanded plots. The changes in conductance between identified level sets (D) are indicated. The conductance changes from each set of transients were analyzed and used to identify the primary conductance change. Source data are available online for this figure.

underly the transient. The initial analysis of the data revealed that the average transient duration, peak conductance, and residual conductance were indistinguishable between WT and KD-RON2 parasites. This suggests that the underlying molecular structure for the transient is generally preserved in the absence of RON2, yet not identically (see below). Likewise, the time dependence of the conductance change is not consistent with a simple model of one on/off channel opening and closing, which predicts one step increase in conductance followed by a step decrease of the same size. Not only are the recordings visually different from this prediction, but the transient relaxes to a residual conductance greater than the initial baseline, followed by a more variable time to return to baseline.

While instructive, comparing the WT transient maximum and quantal step conductances to ultrastructural images of what could be the corresponding structure(s) is difficult. During _T. gondii_ invasion, the only published size of a potential pore is 40 nm in diameter, measured in an image captured by freeze-fracture electron microscopy after the moving junction had formed (Dubremetz, 2007). In this image, the host cell plasma membrane patch that had been in contact with the apical tip of the parasite and is now the presumptive PVM is smooth and continuous, other than the one circular structure. If a pore of this size was present and stayed open during all of invasion, the pore would have been detected in the time-resolved admittance measurements of the entire invasion process (Suss-Toby et al, 1996) but was not.

Furthermore, if a physical cylindrical pore 40 nm in diameter were to form, the membrane conductance would be 34 nS and not the range of 1–6 nS observed here. Using a simple model to convert the peak conductance of the transients observed in our study to the diameter of a single cylindrical pore yields diameters of only 3–10 nm (Fig. EV5). The freeze-fracture image might, however, depict a structure that contains the fused residua of several invasion pores.

## Multiple pores open and close in the host cell membrane during the invasion of _Toxoplasma_

The characteristics of the transient obtained from the high-resolution analysis suggest that multiple poration events occur to increase the permeability of the host cell membrane after rhoptry exocytosis and independent of complete moving junction formation. Detailed analysis of transients revealed a fast increase in the conductance with a slower decay, each composed of multiple "steps in conductance" identified by change point analysis (Fig. 3A). The change point analysis supports the hypothesis that the kinetic processes for creating and removing the conductance pathways for WT and KD-RON2 are similar (rise and fall time distributions Fig. 4A). The identification of abrupt changes in conductance during defined change point mean conductances were further described using PWC analysis where quantal conductance changes of 0.26 and 0.19 nS were identified in the WT and KD-RON2 change point means, respectively. We propose that the observed quantal

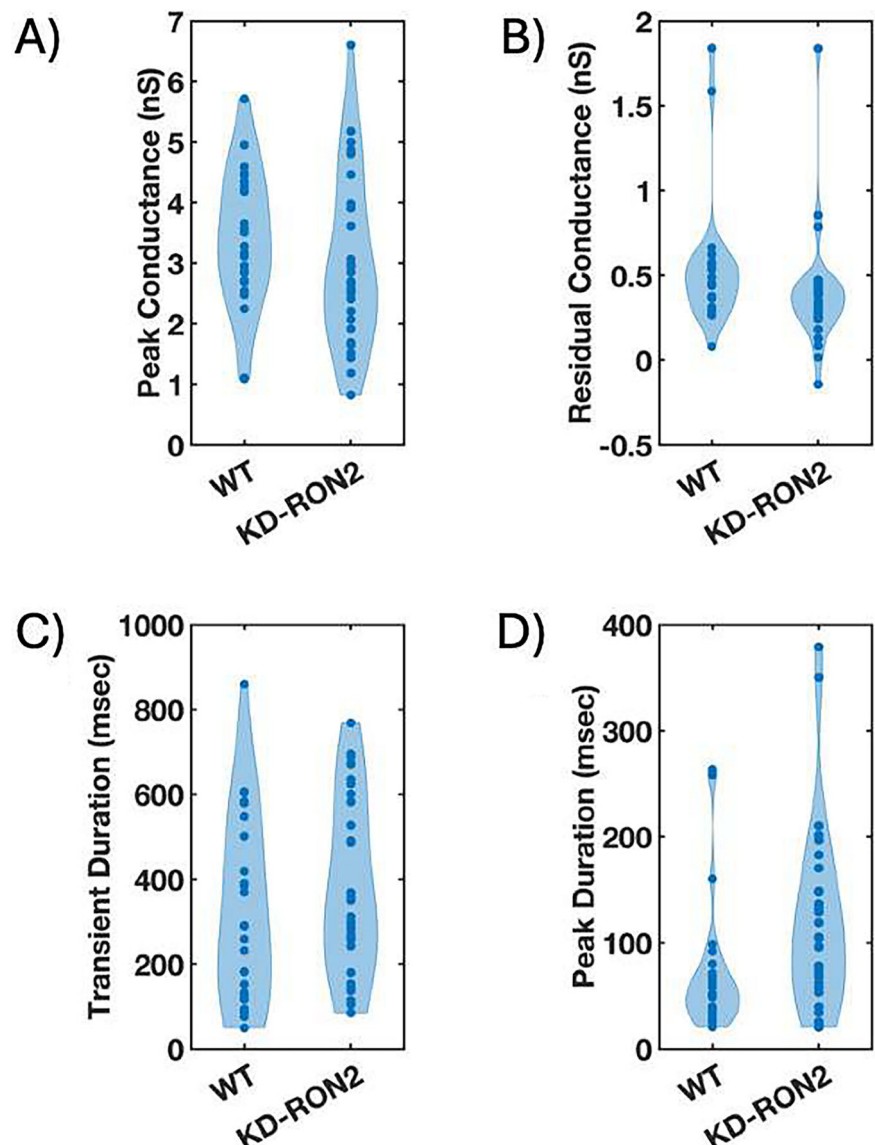

**Figure 4. The conductance transients induced by WT and KD-RON2 tachyzoites differ in peak duration but not in transient duration nor peak or residual conductance.**

(A–D) Violin plots comparing the waveform parameters obtained for WT ($n = 25$) and KD-RON2 parasites ($n = 30$): (A) peak conductance, (B) residual conductance, (C) transient duration, and (D) peak conductance duration. Note, in 4D the peak durations are different; bootstrap differences in the mean, alpha = 0.05; ANOVA, $p = 0.034$ without Box-Cox transformation; following Box-Cox transformation and Welch's $T$-test for unequal variance, $p = 5.69E-17$. Source data are available online for this figure.

conductances arise from invasion pores whose apparent opening results in the observed rising phase (through either incorporation or stimulation by rhoptry content molecules) and whose apparent closing results in the observed falling phase (through occlusion, inactivation or gated closing) together with a flickering process (rapid opening and closing) present throughout the transient. For both parasite lines, the averaged maximum conductance is modeled as the action of 13–16 quantal units acting additively (Fig. EV3A,B).

Alternative explanations for a quantal step size are worth exploring. Multiple steps in conductance are not unusual for proteinaceous pores having subconductance states, such as the voltage-dependent anion channel from Neurospora or rat liver

(Zimmerberg and Parsegian, 1986). The largest component in the distribution of conductance step sizes for VDAC is the main open-closed transition (also ~0.25 nS in salt solutions close to those used here). A single proteinaceous pore model with 13–16 identical sub-conductance units (size, duration, and open and closing properties), is unlikely barring an unusual sequential conductance state transition pattern that mimics the exponential rising and falling time distributions observed using the change point analysis. However, our analysis does not rule out multiple proteinaceous pores with lower numbers of subconductance states whose individual stochastic behavior (subconductance states) and combined behavior (activation) result in the macroscopically observed conductance changes.

 

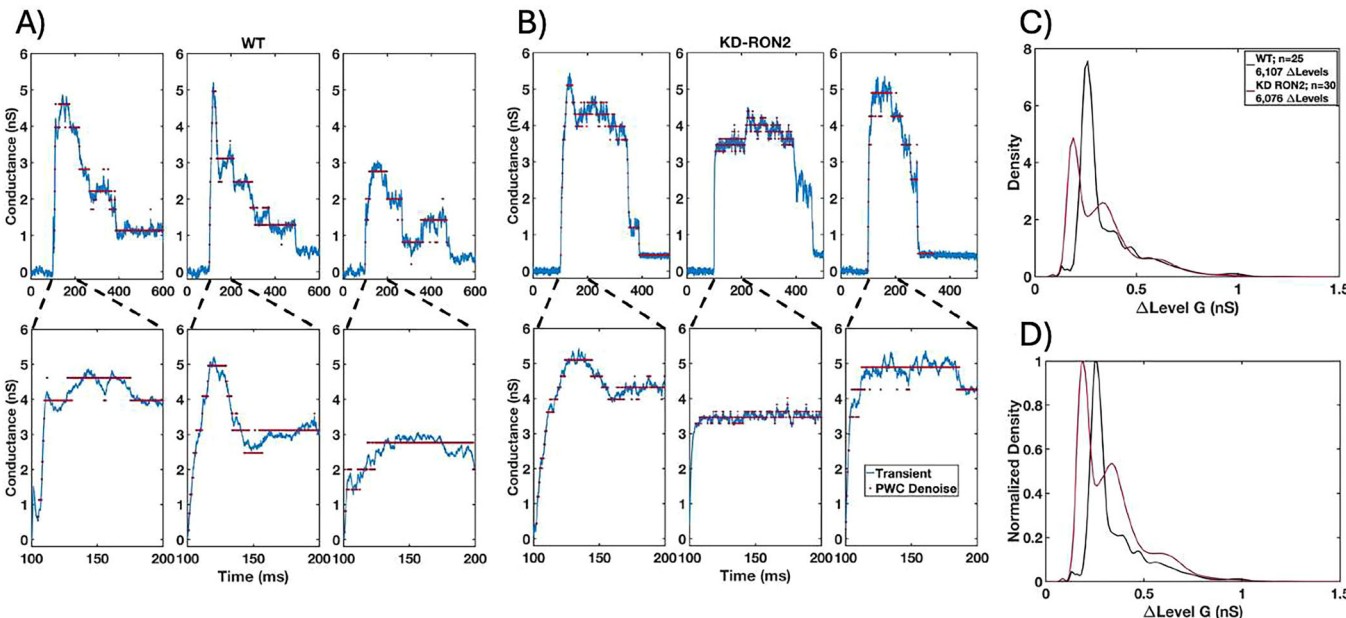

**Figure 5. Evidence for quantal conductance levels from time series analysis of individual transient recordings.**

Piecewise constant (PWC) filtering was used to analyze the time-dependent conductance changes observed during individual transients. Examples of three (**A**) WT-induced and (**B**) KD-RON2 transients analyzed using PWC filtering to calculate the distributions of conductance level changes. (**C**) Density distribution functions of combined transient conductance level changes (ΔLevel G) produced by WT and KD-RON2 strains ($n = 25$ and 30, respectively). (**D**) Peak normalized density of data presented in (**C**). Source data are available online for this figure.

## Similarities and differences between WT and KD-RON2

The lack of significant differences in the peak and residual conductance magnitudes, the time to reach both the peak and the residual conductance, and the transient duration of individual transients (other than the duration of the maximal conductance level) indicates pore formation per se likely occurs irrespective of RON2 and is not interfered with by a moving junction complex that lacks WT levels of RON2, RON4, and RON5 (Lamarque 2014). As documented above, the rising and falling phases are kinetically and distributionally indistinguishable for both WT and KD-RON2 transients, suggesting a RON2-independent insertion and removal process coupled to a RON2-dependent process that influences the lifetime of the maximal conductance state (Fig. 2B). However, the largest quantal conductance component is different between WT (0.26 nS) and KD-RON2 (0.19 nS) parasites. This difference could result from RON2 slightly altering the structure of the invasion pore, either directly or indirectly. For example, this difference could arise from a change in the access resistance to the pore (e.g., altering the charge on the pore mouth). Direct physical measurements of the dimensions and density of the putative adhesion zone between the parasite and the host cell, with and without RON2, could be useful in estimating other forms of the pore access resistance. Another explanation for the difference in the quantal conductance distributions between WT and KD-RON2 parasites can be that depletion of RON2 modifies which rhoptry proteins pass through the pore, how they pass, or their local concentration near the invasion pore. Any of these factors could have an indirect impact on how ions flow through the pore, and might also explain the longer residence time of the KD-RON2 pore indicated by the maximal conductance duration (Fig. 4D). This result is understandable if the

function of the invasion pore is related to the passage of rhoptry cargo through the pore: longer times are needed to move cargo through a smaller pore, and once the cargo has passed through, the invasion pore is no longer needed.

## Known apicomplexan pores and translocons

The detailed electrophysiological and statistical analysis of the invasion pores presented in this paper does not directly address the question of whether these pores serve as the pathway by which rhoptry proteins are translocated into the host cell during invasion. If the pores described here do function in rhoptry protein translocation, they may share molecular features with other known protein translocons (Rapoport 2024). In this study, the conductance transient induced during invasion is best fit by quantal increments of ~0.26 nS, strikingly like the single channel values obtained by direct measurement for the pore-forming protein EXP2 (Garten et al, 2018), a critical component of the well-studied PTEX translocon in malaria parasites (de Koning-Ward et al, 2009; Kalanon et al, 2016). PTEX functions to export proteins across the parasitophorous vacuolar membrane (PVM) and into host red blood cell cytoplasm. The membrane conductance of EXP2 was measured using an on-vacuole patch-clamp method, yielding a functional channel conductance of ~0.2–0.3 nS (Garten et al, 2018).

In *T. gondii*, few pore-forming proteins or potential translocons have been identified. The PVM-associated Myr protein complex functions after invasion in the translocation of secreted dense granule proteins across the PVM into the host cell cytosol (Franco et al, 2016; Cygan et al, 2020). Dense granule proteins GRA17, GRA23, GRA47, and GRA72 are pore-forming proteins that also

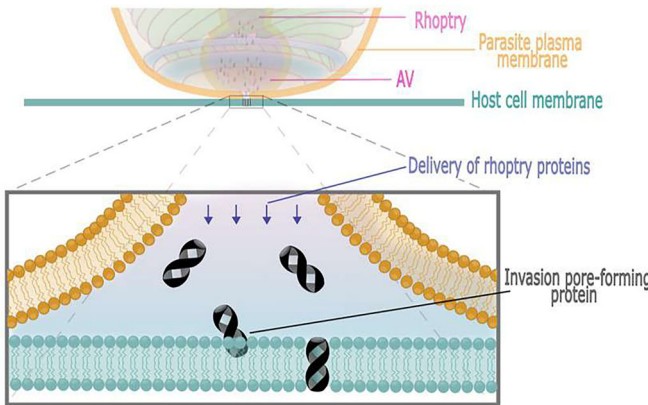

**Figure 6.  Model for invasion pore formation during parasite invasion.**

Invasion pore formation is triggered by rhoptry exocytosis (see companion paper, Male et al, 2025). Electrophysiological analysis supports multiple rather than single pore formation on the host cell membrane. The model depicted in this figure has pore-forming proteins emerging from the mouth of the fusion pore between the AV and parasite plasma membrane to incorporate into the underlying proximal host cell membrane. The parasite plasma membrane is orange and the host cell plasma membrane is green. While we think that it is likely that rhoptry proteins, perhaps after mixing with AV-resident proteins, have pore-forming activities, we cannot exclude the possibility that rhoptry proteins collaborate with resident host cell plasma membrane proteins to form the invasion pores.

appear to function post-invasion in nutrient transport across the PVM (Gold et al, 2015; Bitew et al, 2024), although a role for these proteins in protein translocation cannot be ruled out. Intriguingly, *Plasmodium* EXP2 can rescue loss-of-function phenotypes in *Toxoplasma* lacking GRA17 (Gold et al, 2015). *Toxoplasma* also encodes two perforin-like proteins, PLP1 and PLP2 (reviewed in Carruthers, Ann Rev Microbiol 2024). PLP2 is only expressed in the sexual stages (Kafsack et al, 2009), but PLP1 forms pores in the PVM and host cell membrane during tachyzoite egress from the host cell (Roiko and Carruthers, 2013). PLP1 pore-forming activity is pH-dependent, with higher activity in acidic compared to neutral conditions (Roiko and Carruthers, 2013). Invasion assays conducted in low pH conditions result in increased levels of parasite attachment and invasion, due to increased microneme secretion, and also increased levels of host cell wounding, potentially due to PLP1 (Roiko and Carruthers, 2013). Based on the structure of PLP1 (Ni et al, 2018), it seems unlikely it can function as a protein translocon. To help in identifying proteins responsible for pore formation, bioinformatic analyses, including screening for pore-forming structural motifs like amphipathic alpha helices and cell-penetrating peptides, combined with protein structure prediction methodologies such as AlphaFold, are currently underway. In addition to the above pore formation motifs, protein translocon features are also under consideration (Rapoport, 2024).

The mechanism underlying the rapid closure of the invasion pore reported here is also a mysterious phenomenon. While not directly related to parasite invasion, membrane resealing in model systems of membrane repair is well documented, occurring on time scales from ~1 to 10s of seconds. (Steinhardt et al, 1994; Togo et al, 1999; Terasaki et al, 1997; Bansal et al, 2003; Humphrey et al,

2012; Klenow et al, 2021). Closing of the invasion pores may reflect (1) a host cell response to repair the sudden change in permeability through rapid endocytosis (Corrotte et al, 2020), (2) blebbing of the plasma membrane to remove the pores following their insertion (Jimenez et al, 2014), or (3) gated closure of the invasion pore once cargo is delivered, a feature of many protein translocons (Rapoport 2024). The recruitment of host cell proteins implicated in membrane repair and resealing, such as ESCRT proteins or dysferlin, may be a useful approach to test the hypothesis that pore closure involves a host cell response. It will also be important to determine to what extent material passing through the pore (e.g., ROP proteins) contributes to the decrease in ion flux through simple occlusion of the pore. Finally, the existence of residual conductance implies a further host cell plasma membrane phenomenon requiring additional studies. While outside the scope of this paper, future biophysical study is also needed to evaluate possible effects of conoid extrusion on invasion pore insertion or function; mutants that interrupt the apical polar ring and invasion but retain rhoptry secretion are of interest (Ren et al, 2024).

## Conclusions

In this study, we describe a transient permeability pathway in the host cell membrane during *Toxoplasma* invasion. The electrophysiology data are best explained by synchronous creation of multiple pores by material secreted from the rhoptries into the host cell plasma membrane (Fig. 6). The function of these pores is unknown but their similar size to protein translocons and their appearance in the sequence of invasion (after rhoptry exocytosis and before moving junction formation) suggest the hypothesis that this poration creates a pathway for delivery of secreted rhoptry proteins into the host cell cytoplasm.

## Methods

**Reagents and tools table**

| Reagent/resource | Reference or source | Identifier or catalog number |
|---|---|---|
| **Experimental models** | | |
| CCD-1112Sk; foreskin fibroblast (*Homo sapiens*) | ATCC; authenticated by STR profiling | CRL-2429 |
| COS1; kidney fibroblast (*Cercopithecus aethiops*) | ATCC | CRL-1650 |
| *T. gondii*: strain RH | Gift from Dr. Alan Sher, NIH | |
| *T. gondii*: strain RON2 cKD | Lamarque et al, 2014 | |
| **Antibodies** | | |
| Mouse monoclonal anti-SAG1 (clone DG52) | Gift from Dr. David Sibley Origin: Burg et al, 1988 | |
| Alexa Fluor 647 Antibody Labeling Kit | Thermo Fisher Scientific | A20186 |
| **Chemicals, enzymes and other reagents** | | |
| Molecular Probes, Fluo-4, AM, cell permeant | Thermo Fisher Scientific | F14201 |

| Reagent/resource | Reference or source | Identifier or catalog number |
|---|---|---|
| Molecular Probes, Powerload Concentrate | Thermo Fisher Scientific | P10020 |
| Sigma Aldrich Calcium Ionophore A23187 | Thermo Fisher Scientific | 50-176-5967 |
| Anhydrotetracycline | Takara Bio | 631310 |
| **Software** | | |
| NIS Elements v. 5.11 | Nikon | |
| FIJI | https://imagej.net/ | |
| MATLAB | MathWorks https://www.mathworks.com | |
| PeakCaller | Artimovich et al, 2017 https://hussmanautism.org/resources/software/ | |
| R, version 4.0.1 | R Foundation for Statistical Computing http://www.R-project.org | |
| Prism, version 10.4.1 | GraphPad http://www.graphpad.com | |
| Clampfit 11.2 | Molecular Devices | |
| **Other** | | |
| ibidi μ-Slide VI 0.4 | Thermo Fisher Scientific | 50-305-784 |
| 35 mm DT dish | Bioptechs | |
| 1.5-mm thick-wall borosilicate glass capillaries | Sutter Instruments | |

## Culture of fibroblast-like COS1 cells

Fibroblast-like COS1 cells (ATCC CRL-1650) were cultured in complete Dulbecco's modified Eagle's medium (DMEM supplemented with high glucose, 200 mM Glutamax, 1 mM sodium pyruvate, ThermoFisher Scientific, Cat#10569, Waltham, MA) with 10% Fetal Bovine Serum Premium Select (R&D Systems, Cat#S11550, Minneapolis, MN) and primocin 100 μg/mL (Invivo-Gen, Cat# ant-pm-2, San Diego, CA) at 37 °C under 5% $CO_2$. Cells were rinsed twice with DPBS without both $CaCl_2$ and $MgCl_2$ (Thermo Fisher Scientific, Cat#14190, Waltham, MA). Cells were released from the flask using a 5-min incubation with 0.25% trypsin containing 0.913 mM EDTA (GIBCO, Cat#25200-056, Waltham, MA). Trypsinized cells were centrifuged at 193×$g$, 5 min, room temperature (RT), in complete DMEM, using an Allegra X-22R centrifuge with Beckman Coulter rotor. Cells were resuspended in complete medium, and 1 mL containing 40,000 cells was seeded onto a 35 mm DT dish (Bioptechs, Butler, PA) for 60 min at 37 °C under 5% $CO_2$. Before starting electrophysiology experiments, DMEM medium was replaced by live cell imaging solution (LCIS) containing 155 mM NaCl, 3 mM KCl, 2 mM $CaCl_2$, 1 mM $MgCl_2$, 3 mM $NaH_2PO_4$, 10 mM HEPES, and 20 mM Glucose (final pH 7.4) or low-calcium LCIS containing 155 mM NaCl, 3 mM KCl, 0.1 mM $CaCl_2$, 2.9 mM $MgCl_2$, 3 mM $NaH_2PO_4$, 10 mM HEPES, and 20 mM Glucose (final pH 7.4).

## Culture of human foreskin fibroblasts (HFFs)

HFFs were maintained in Dulbecco's Modified Eagle Medium (DMEM supplemented with high glucose, 200 mM Glutamax, 1 mM sodium pyruvate, ThermoFisher Scientific, Cat#10569, Waltham, MA) with 10% v/v heat-inactivated fetal bovine serum (FBS) (R&D Systems, Cat#S11550), 50 U/mL Penicillin-Streptomycin (Thermo Fisher Scientific, Cat#15140122, Waltham, MA) and 25 μg/mL gentamicin (GIBCO, Cat#15750, Waltham, MA). Prior to parasite passage, HFF medium was replaced with fresh culture medium (DMEM containing 10% v/v heat-inactivated FBS, 50 U/mL Penicillin-Streptomycin and 25 μg/mL gentamicin.

## *Toxoplasma gondii* culture and isolation

*T. gondii* RH (hereafter designated WT) and KD-RON2 (Lamarque et al, 2014) strains were passaged in human foreskin fibroblasts (HFFs) (ATCC CRL-1634) grown in DMEM supplemented with 10% heat-inactivated FBS (R&D Systems, Cat#S11550, Minneapolis, MN), 50 U/mL Penicillin-Streptomycin (Thermo Fisher Scientific, Cat#15140122, Waltham, MA), and 25 μg/mL gentamicin (GIBCO, Cat#15750, Waltham, MA) at 37 °C under 5% $CO_2$. *T. gondii* tachyzoites were isolated from infected monolayers at the large vacuole stage. Freshly released tachyzoites were removed and discarded before collecting infected cells from the culture flask by replacing culture medium with 4 mL of Endo buffer containing 89.4 mM KOH, 44.7 mM $H_2SO_4$, 10 mM $MgSO_4$, 106 mM sucrose, 5 mM glucose, 20 mM Tris-$H_2SO_4$, and 3.5 mg/mL BSA (final pH 8.20, adjusted using KOH) (Endo et al, 1987). Attached cells and parasites were collected by scraping and resuspended in Endo solution. Collected infected cells were passed three times through a 25G needle and filtered through a 5-μm cellulose acetate filter (Sartorius, Cat#S7594-FMOSK, Göttingen, Germany). Isolated parasites were washed three times with Endo solution using a 10 min centrifugation at 400×$g$ and resuspended in 50 μL Endo solution for electrophysiological experiments. Parasites were used within 2 h following isolation and were protected from light exposure.

## Calcium transient assay

### *T. gondii* culture and isolation

For calcium transient experiments, "untreated" KD-RON2 parasites were grown in medium containing 0.07% ethanol for 48 h prior to experiments for consistency with all other untreated controls reported in (Male et al, 2025). Isolated parasites were resuspended in anti-SAG1 antibody (monoclonal antibody DG52, a generous gift from Dr. David Sibley, 1/20 dilution of a 0.2 mg/mL stock) conjugated to AlexaFluor 647 (Alexa Fluor™ 647 Antibody Labeling Kit; Molecular Probes, Eugene, OR) in Endo buffer for 30 min at RT, then centrifuged at 1000 × $g$ for 2 min, and resuspended in Endo buffer at $3 × 10^7$ parasites/mL.

### Cell labeling

HFFs were seeded in three chambers of an ibidi μ-Slide VI 0.4 (ibidi GmbH, Gräfelfing, Germany) for 2 h at 37 °C (with 5% $CO_2$ and humidity) before indicator loading with 5 μM Fluo-4 AM (Invitrogen, Waltham, MA) with 1% PowerLoad Concentrate, 100X (Invitrogen, Waltham, MA) in LCIS for 90 min at RT.

## Imaging and data analysis

Experiments were carried out at 35–36 °C. Fluo-4 labeled cells were washed 1× with Endo buffer before allowing anti-SAG1 pre-labeled parasites to settle on the cells for 10 min. The buffer was then exchanged with pre-warmed (35–36 °C) invasion-permissive LCIS to capture calcium transients and invasion events. Imaging was carried out on a Nikon Eclipse TE300 widefield epifluorescence microscope (Nikon Instruments, Melville, NY) using a 60× PlanApo λ objective (NA 1.4). 1020 × 1020-pixel images were captured using an iXon 885 EMCCD camera (0.22 μm/pixel, Andor Technology, Belfast, Ireland) set to trigger mode, with exposure time of 39 ms, no binning, 30 MHz readout frequency, 3.8× conversion gain, and 300 EM gain. Parasite-induced perforation of host cells resulted in calcium transients, and subsequent invasion events were observed by near-simultaneous excitation of Fluo-4 (490 nm) and AlexaFluor 647 (635 nm) using a pE-4000 LED illumination system (CoolLED, Andover, England) through rapid excitation switching triggered by the NIS Elements Illumination Sequence module (Nikon Instruments, Melville, NY). Hardware was driven by NIS Elements v. 5.11 software (Nikon Instruments, Melville, NY). Calcium transient and invasion events were quantified as described elsewhere (Male et al, 2025) across three biological replicates, each consisting of three technical replicates.

## Electrophysiology

Electrophysiological experiments were performed with COS1 cells seeded onto a 35 mm DT dish containing LCIS, the external recording solution. The temperature during the experiment was kept at 37 °C. The pipette solution for whole cell recordings contained 122 mM KCl, 2 mM MgCl$_2$, 11 mM EGTA, 1 mM CaCl$_2$, 5 mM HEPES (Final pH 7.26, adjusted with KOH). Patch pipettes (~3 MΩ resistance) were fabricated from 1.5-mm thick-wall borosilicate glass capillaries (P1000, Sutter Instruments, Novato, CA). Electrophysiological parameters of each patched cell were monitored with an amplifier (AxoPatch200b, Molecular Devices, San Jose, CA) in the voltage-clamp mode. The output current was filtered using the internal 100 kHz Lowpass Bessel filter included in the amplifier, and an external 5 kHz low-pass 8-pole Bessel filter (Model 900 CT/9 L8L, Frequency Devices Inc, Haverhill, MA). A −60 mV holding potential was applied to monitor current changes due to the interaction of each strain of *Toxoplasma* tachyzoites with COS1 cells. Current was recorded for a maximum of 15 min per cell, digitized at 100 μs (Axon Digidata 1550B and its associated software package Axopatch, Molecular Devices, San Jose, CA). Time resolution of <200 μs was empirically measured, matching the expected value determined by the interaction of the external 5 kHz filter and the whole-cell compensation circuitry of the amplifier. Conductance was calculated using Ohm's law adjusted using the mean pipette access resistance, $R_A$ ($n = 40$, $R_A = 2.88/0.34$ MΩ mean/sem); with our internal and external solutions, the measured reversal potential of the COS1 cells was ~0 mV. Data analysis was performed offline (Clampfit 11.2, Molecular Devices, San Jose, CA and MATLAB R2022b, MathWorks, Natick, MA).

## Tachyzoite delivery

Delivery pipettes were fabricated from 1.5-mm thick-wall borosilicate glass capillaries using a pipette puller (P-80/PC, Sutter Instruments, Novato, CA). Delivery pipette inner diameters ranged from 10 to 20 μm, which was large enough to allow tachyzoites to pass through without damage. Delivery pipettes were bent (<45°) over the flame of a small lighter to facilitate a vertical and smooth delivery of parasites onto the surface of a target cell. Parasites were delivered adjacent to the patched cell using a microinjector (FemtoJet® 4i, Eppendorf, Cat#5252000021D, Enfield, CT) set at ~5 hPa.

## Optical visualization of invasion and image analysis

Imaging was initiated immediately after a stable, whole-cell configuration was achieved. Parasite invasion was monitored on an inverted microscope (Axiovert 200, Zeiss, Oberkochen, Germany) equipped with a 63 × 1.4 N.A. objective, differential interference contrast (DIC) optics, and a sCMOS camera (12.2 frames per second, 1501M-GE, ThorLab, Newton, NJ). Imaging proceeded for 15 min or until disruption of the giga-ohm seal required for whole cell electrophysiology, whichever occurred first. Offline, image files were screened for the presence or absence of parasite invasion, identified by DIC as a visible constriction of tachyzoites at the COS1 cell plasma membrane as they entered the cell. The number of parasites delivered onto the cell surface, as well as the number of invasions were counted to determine a ratio of invasion events per parasite.

## Data analysis for the electrophysiological recordings

Electrophysiological data were processed using MATLAB. Digitized data were converted into a current and voltage data array using abfload (Hentschke, v1.4.0.0, MATLAB Central File Exchange). Conductance and current plots were generated from the raw data following median filtering. MatLab median filter function, medflt1, was applied to remove noise spikes from the raw data to facilitate identification of biologically relevant transients. The choice of the median filter (9 points) was established empirically, balancing rejection of noise spikes, with minimal reduction in the amplitude of the parasite-induced transient. Typically, the noise spikes are significantly shorter than biologically relevant spikes with typical durations of ~0.4 ms. To identify the biologically relevant transients, several criteria were used, including changes in the moving variance, first derivative, and manual inspection. The hand-curated selection of data was implemented in MATLAB scripts to extract the transients. To identify transient and peak conductance starting and ending times, sequence change point analysis (MatLab findchangepts) was used to identify the times of mean conductance changes that persisted longer than 10 ms or 100 samples (100 μs each) (Fig. 3A). Change point analysis identifies a change in signal statistics with time, for example the mean, of the whole cell conductance signal. As the number of change points existing in the conductance data was unknown, a range of fixed values were added to the residual error (a penalty term linear with the number of change points to compensate for the decrease in residual error that each additional change point introduced), until a plateau in the residual error was observed. The change point parameter set associated with stabilizing the residual error was used to calculate global features of the conductance transient: peak and residual conductance magnitudes, time to reach both the peak and residual conductance, and transient and peak duration. The peak is calculated from the magnitude of the maximum change-point conductance value, while the peak duration is the difference in time between the two change points defining the peak conductance. The

 

transient duration is calculated from the difference between the first and last change points. The residual conductance is the difference between the mean of the last 50 ms and the mean of the first 95 ms in the 1 s time series capturing each transient, where the first detected change in conductance for each spike is set at 100 ms. Time to reach the peak conductance is defined by the time between the start of the transient and the start of the maximum change point mean statistic, and the time to reach the residual conductance is defined by the time between the end of the maximum change point mean and the last change point. Modeling the total resistance as the sum of the pore $\left(R_{pore} = \frac{\rho * L}{\pi * r^2}\right)$ and twice the pore access $\left(R_{access} = \frac{\rho}{4 * r}\right)$ resistances, calculation of the radius $(r)$ of a cylindrical pore (length $L$, solution resistivity $\rho$) with maximum conductance $G_{max}$ of each transient is

$$r = \left(G_{max} * \frac{\rho}{4}\right) * \left(1 + \left(1 + \frac{4^2 * L}{G * \rho * \pi}\right)^{0.5}\right.$$

## Statistical analyses

Comparisons between the global features of the conductance transients observed in WT and KD-RON2 (peak and residual conductance, and transient duration) were evaluated using one-way analysis of variance (ANOVA) following both tests for normality (Lilliefors Test) and homogeneity of variance (Levene's Test). The residual conductance distributions were not normal, displaying long-tails; one-way ANOVA of Box-Cox transformed data was used following the removal of one negative value in the KD-RON2 data set. To validate comparisons relying on distributional assumptions (one-way ANOVA), the bootstrap with 100,000 replicates was used to calculate 95% confidence intervals (CIs) at an alpha level of 0.05 for differences between data set means. CIs bracketing zero were consistent with no evidence for a significant difference between data sets. Rise and falling time distributions were compared using the two-sample Kolmogorov–Smirnov test. Rejection of the null hypothesis was set at $p < 0.05$ for ANOVA, Lilliefors, Levene's, and Kolmogorov–Smirnov tests. Blinding was not done. Sample size estimation was not performed.

Two mixture model approaches were used to analyze distributions of data to provide a robust assessment: (1) fitting the empirical cumulative distribution function (eCDF) to a parametric model consisting of a weighted sum of normal or log-normal distributions and (2) applying a Gaussian mixture distribution model using both supervised (MatLab GMModel, iterative expectation maximization) and unsupervised (Wallace and Dowe, 2000 as coded by Statovic V 0.81, minimum message length) clustering algorithms. Graphs showing 95% confidence interval shadings were prepared using the Gramm data visualization toolbox for MATLAB (V 2.27.1 by Pierre Morel). All statistical tests were two-sided unless indicated otherwise.

## Determination of clustered conductance values

Piecewise constant (PWC) filtering was used to analyze the time-dependent conductance changes observed during a transient (Little and Jones, 2011; implemented in MATLAB using pwc_cluster.m). PWC filtering is an unsupervised cluster analysis amenable to one-dimensional time series that optimizes membership within

identified conductance levels (clusters). PWC filtering preserves sharp boundaries and other characteristics of rapid transitions ("jumps") in the data, and it does not require that the number of clusters be specified because the analysis is unsupervised. Here, a conductance level or state is hypothesized to be PWC with variable magnitude and duration depending upon the number and lifetime of individual conductance units present. Like change point analysis, removing noise from PWC signals is a signal processing optimization problem with many established iterative procedures. Here, PWC filtering was applied using a clustering algorithm where each data point is shifted towards the highest density (mean-shift) within a fixed distance centered at the data point (hard kernel) (Little and Jones, 2011). In addition, the number of uniquely sized conductance levels following optimization is related to the amount of filtering and is set by an additional parameter called a "tuning hyperparameter". Since the amount of filtering can be adjusted using this tuning hyperparameter, the PWC conductance levels are sensitive to the magnitude of this parameter, and there is a risk of fitting the noise if the parameter is too small. To avoid fitting the noise, an optimization procedure that identified the noise level of the baseline was adopted. The ~100 ms baseline prior to the start of a transient (as above, it is set by us at 100 ms) was assumed to represent a PWC signal with noise properties associated with the whole cell configuration and that the noise present in the baseline is present and constant for the duration of the transient. The minimum constant conductance level change identified using the PWC filtering was set to be greater than the 99% confidence interval of the noise present in the constant baseline. The tuning hyperparameter was iteratively adjusted over a range until the resulting level changes were greater than the minimum constant conductance level change identified in the baseline noise. The hyperparameter satisfying the constraint defined by the baseline noise was considered optimal filtering for the transient. Since the noise varied between experiments, the resulting level changes were pooled from individually processed conductance transients.

## Testing for quantal conductance changes using the differences between adjacent clustered values

The magnitude of the level changes, obtained by calculating the absolute value of the differences between identified PWC levels present in the conductance transient time series, constitutes the data sets whose distributional properties were analyzed as described above. To create the set of level changes, the PWC representation of the conductance time series was numerically differentiated by taking first-order differences between the time series ($t_i$-$t_{(i-1)}$). Zero differences were dropped; only the positive and negative changes between conductance levels were analyzed. These changes represent the 'jumps' between levels at the specified time, where the filtering defined a change in the cluster set associated with the new level. Except for the sign, the positive and negative level changes had symmetric and equal distributions, justifying working with the absolute values of the level changes. If the distribution of level changes was uniform across all possible conductance differences (all possible conductance steps are represented in the data set), then there would be no evidence for a dominant peak or mode. However, if the distribution has one or more well-defined peaks, then these peaks can be analyzed as a weighted sum of individual Gaussian distributions using mixture models to identify the peak

with the greatest contribution to the overall distribution. Distributional peaks were visualized using a non-parametric density approximation prior to Gaussian mixture model analysis for identification and statistical evaluation of the peak properties, as described above. Comparisons between modes of multi-modal distributions were made by first isolating sub-distributions of data using the 95% CI around the Gaussian mixture model identified peaks, followed by pair-wise and group analyses (ANOVA and Kruskal–Wallis) and subsequent post-hoc multiple comparisons testing using Tukey's honestly significant difference or Dunn's test, respectively.

## Data availability

This study includes no data deposited in external repositories.

The source data of this paper are collected in the following database record: biostudies:S-SCDT-10_1038-S44319-025-00565-8.

## Peer review information

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

## Acknowledgements

Transgenic parasites were kindly gifted from Dr. Maryse Lebrun from Université Montpellier, whom we thank for useful discussions. This work was supported by the intramural program of the National Institutes of Health (JZ) and US Public Health Service grant U01AI169067 (GEW). YK was supported in part by a Japan Society for the Promotion of Science fellowship from 2021 to 2023.

## Author contributions

**Yuto Kegawa**: Conceptualization; Resources; Data curation; Formal analysis; Validation; Investigation; Visualization; Methodology; Writing—original draft; Writing—review and editing. **Frances Male**: Conceptualization; Resources; Data curation; Formal analysis; Validation; Investigation; Visualization; Methodology; Writing—original draft; Writing—review and editing. **Irene Jiménez-Munguía**: Conceptualization; Resources; Data curation; Validation; Investigation; Visualization; Methodology; Writing—original draft; Writing—review and editing. **Paul S Blank**: Conceptualization; Data curation; Software; Formal analysis; Visualization; Methodology; Writing—original draft; Writing—review and editing. **Elena Mekhedov**: Investigation; Methodology; Writing—original draft; Writing—review and editing. **Gary E Ward**: Conceptualization; Resources; Supervision; Validation; Writing—original draft; Writing—review and editing. **Joshua Zimmerberg**: Conceptualization; Resources; Supervision; Funding acquisition; Validation; Investigation; Writing—original draft; Project administration; Writing—review and editing.

Source data underlying figure panels in this paper may have individual authorship assigned. Where available, figure panel/source data authorship is listed in the following database record: biostudies:S-SCDT-10_1038-S44319-025-00565-8.

## Funding

## Disclosure and competing interests statement

The authors declare no competing interests.

# Expanded View Figures

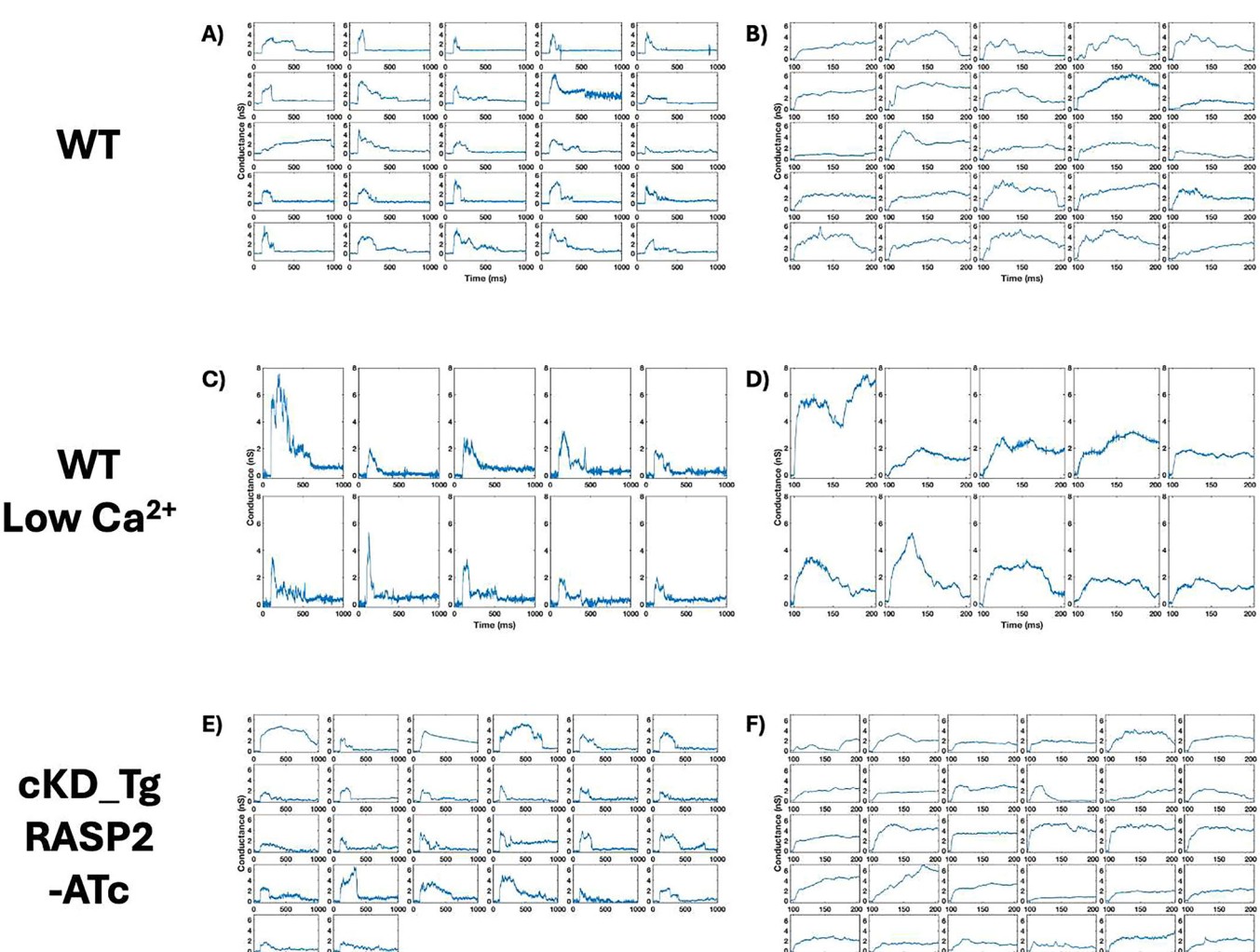

**Figure EV1. Galleries of conductance transients induced by WT and cKD_TgRASP2 -ATc parasites.**

(A) Gallery of 25 WT parasite conductance transients calculated from recorded current measured using −60 mV holding potential in an external buffer containing 2.0 mM $CaCl_2$. The initial 100 ms of baseline is plotted prior to the detection of the transient. (B) Gallery of 25 WT parasite conductance transients calculated from recorded current measured using −60 mV holding potential in an external buffer containing 2.0 mM $CaCl_2$. The initial 5 ms of baseline prior to the detection of the transient and the initial 105 ms of each recorded transient are plotted. (C) Gallery of ten WT parasite conductance transients in low external calcium, calculated from recorded current measured using −60 mV holding potential in an external buffer containing 0.1 mM $CaCl_2$. The initial 100 ms of baseline is plotted prior to the detection of the transient. (D) Gallery of 10 WT parasite conductance transients in low external calcium, calculated from recorded current measured using −60 mV holding potential in an external buffer containing 0.1 mM $CaCl_2$. The initial 5 ms of baseline prior to the detection of the transient and the initial 105 ms of each recorded transient are plotted. (E) Gallery of 26 cKD_TgRASP2 -ATc parasite conductance transients calculated from recorded current measured using −60 mV holding potential in an external buffer containing 2.0 mM $CaCl_2$. The initial 100 ms of baseline is plotted prior to the detection of the transient. Because no ATc or solvent is applied, aside from the genetic alteration of the parasite, no change in the protein complement of this parasite line compared to WT is expected. (F) Gallery of 26 cKD_TgRASP2 -ATc conductance transients calculated from recorded current measured using −60 mV holding potential, showing the initial 105 ms of the transients. 5 ms of baseline is included prior to detection of the transient. Because no ATc or solvent is applied, aside from the genetic alteration of the parasite, no change in the protein complement of this parasite line compared to WT is expected.

 

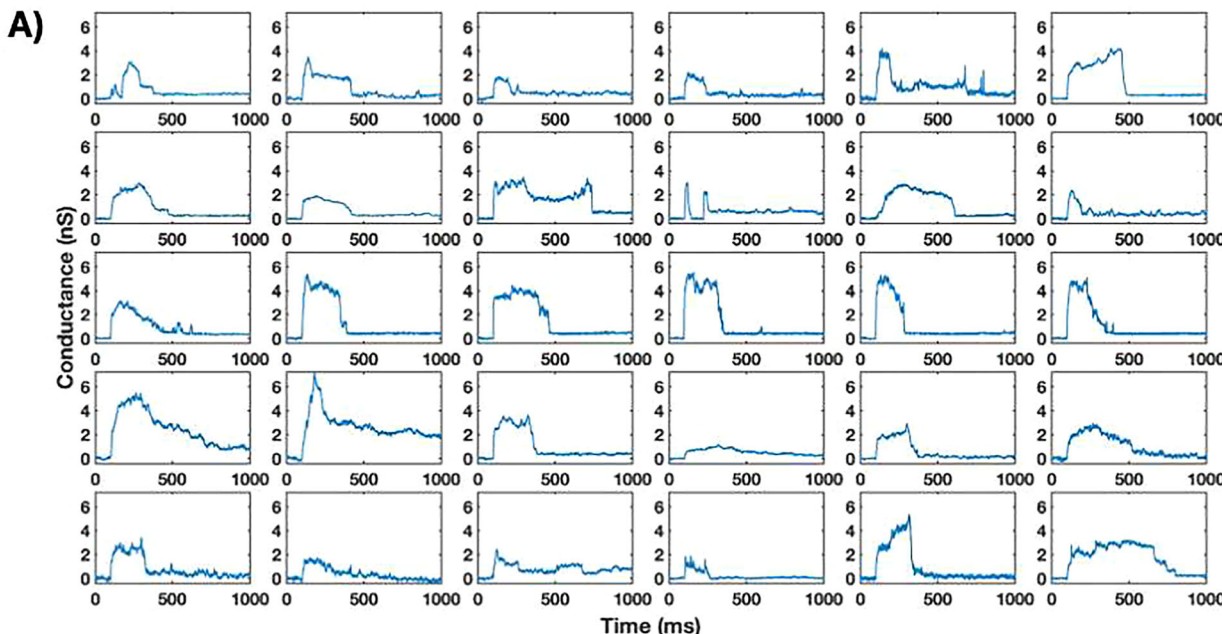

# KD RON2

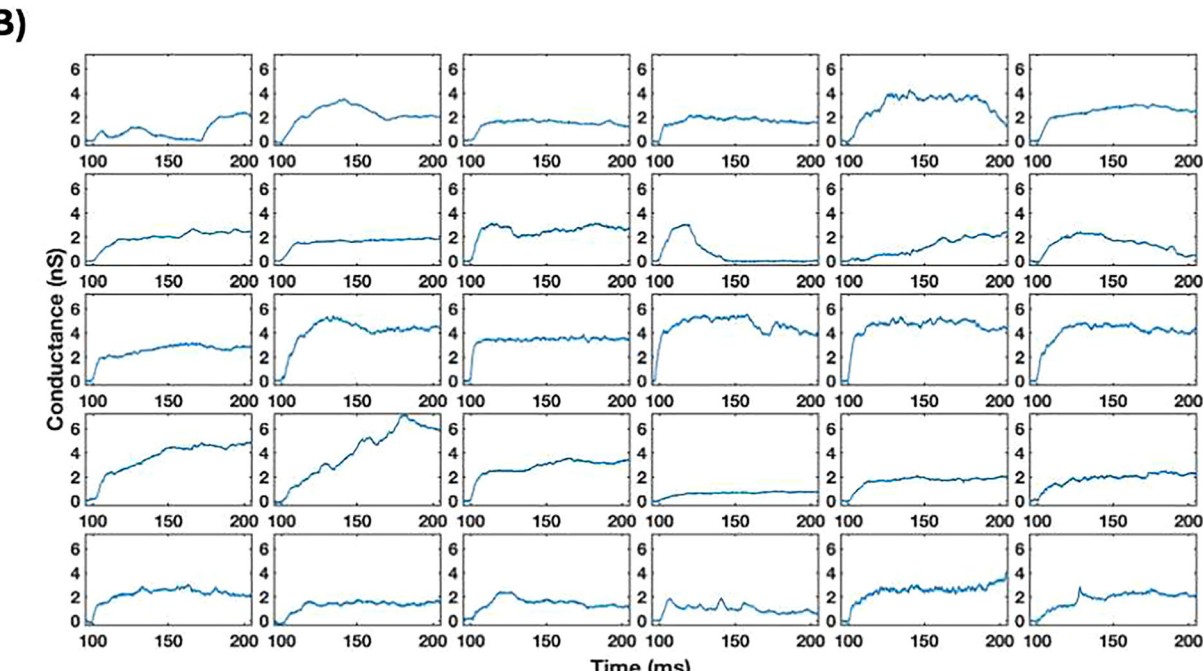

**Figure EV2.  Galleries of conductance transients induced by KD-RON2 parasites.**

(**A**) Gallery of 30 KD-RON2 parasite conductance transients calculated from recorded current measured using −60 mV holding potential in an external buffer containing 2.0 mM $CaCl_2$. The initial 100 ms of baseline is plotted prior to the detection of the transient. (**B**) Gallery of 30 KD-RON2 parasite conductance transients calculated from recorded current measured using −60 mV holding potential, showing the initial 105 ms of the transients in an external buffer containing 2.0 mM $CaCl_2$. The initial 5 ms of baseline is plotted prior to the detection of the transient.

   

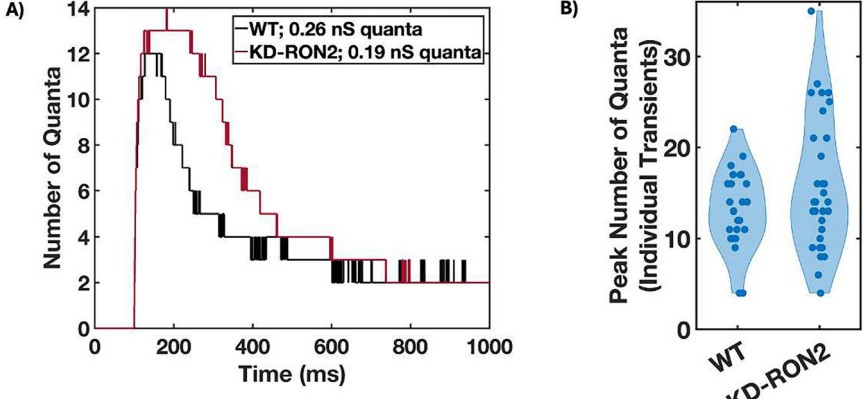

**Figure EV3. The same number of quantal units contributes to the conductance maxima of WT and KD-RON2 transients.**

(A) Transformation of the mean transient conductance (Fig. 3A) using the peak quantal sizes for WT (0.26 nS) and KD-RON2 (0.19 nS), rounded to the nearest integer.
(B) Violin plots of the number of quantal units at the maximum conductance of WT and KD-RON2 transients ($n = 25$ and 30, respectively).

 

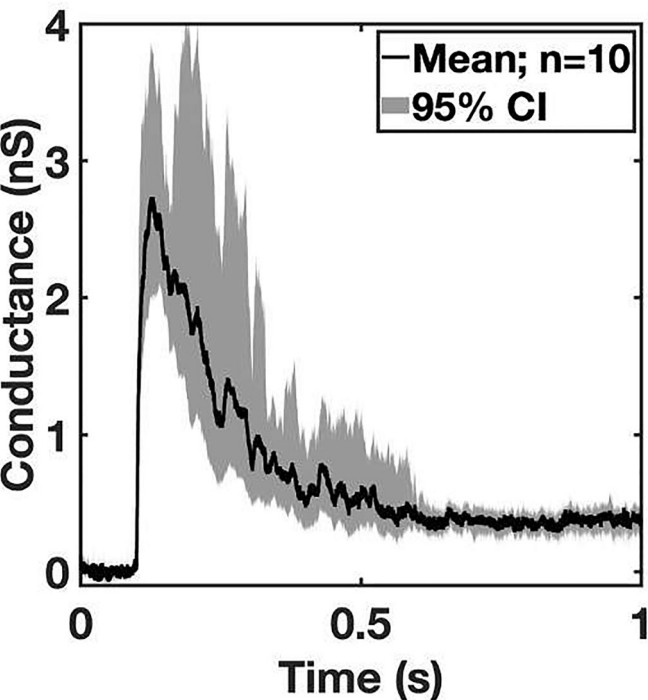

**Figure EV4. Transients induced by WT parasites in low external calcium concentration (0.1 mM CaCl₂) have similar properties to WT transients induced in regular LCIS (2.0 mM CaCl₂; Fig. 3A).**

The average waveform of low-calcium transients displays the characteristic fast rise to peak conductance and slower recovery to a new baseline ($n = 10$). The confidence interval (CI) is calculated for each point along the averaged transient.

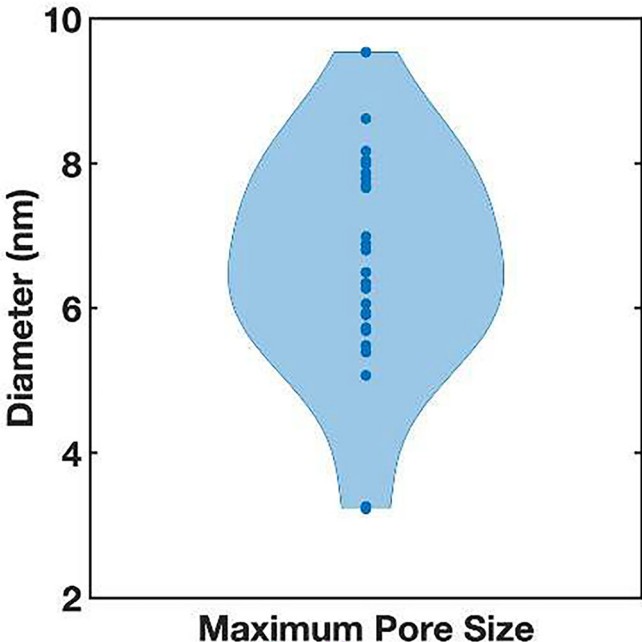

**Figure EV5.** Violin plot of pore diameters calculated from individual WT transient ($n = 25$) maximum conductance using a model for a cylindrical pore (see methods).

 