## [Peer Review File · EMBO Reports]

The invasion pore induced by *Toxoplasma gondii*

Yuto Kegawa, Frances Male, Irene Jimenez Munguia, Paul Blank, Elena Mekhedov, Gary Ward, and Joshua Zimmerberg

Corresponding author: Joshua Zimmerberg (zimmerbj@mail.nih.gov)

Review Timeline:

Submission Date:	29th Nov 24
Editorial Decision:	28th Jan 25
Revision Received:	10th May 25
Editorial Decision:	30th Jun 25
Revision Received:	5th Aug 25
Accepted:	6th Aug 25

Transaction Report:

Dear Dr. Zimmerberg

Thank you for the submission of your research manuscript to our journal. I apologize for the delay in handling your manuscript. We have meanwhile received two referee reports on it and I decided to proceed with these (copied below).

As you will see, the referees acknowledge that the findings are interesting and that the conclusions are overall supported by the data presented but they also raise a number of concerns. Referee 1 has a number of suggestions on testing or discussing the potential contribution of mechanical forces, e.g., by the conoid, and the involvement of membrane repair mechanisms. Mechanistic insight into the nature of the pore is not required for publication at EMBO Reports, but I note that referee 2 suggests experiments that might help to resolve the question whether the pore is proteinaceous or not.

Given the constructive comments, we would like to invite you to revise your manuscript with the understanding that the referee concerns (as detailed above and in their reports) must be fully addressed and their suggestions taken on board. Please address all referee concerns in a complete point-by-point response. Acceptance of the manuscript will depend on a positive outcome of a second round of review. It is EMBO Reports policy to allow a single round of revision only and acceptance or rejection of the manuscript will therefore depend on the completeness of your responses included in the next, final version of the manuscript.

We realize that it is difficult to revise to a specific deadline. In the interest of protecting the conceptual advance provided by the work, we recommend a revision within 3 months (April 28). Please discuss the revision progress ahead of this time with the editor if you require more time to complete the revisions.

I am also happy to discuss the revision further via e-mail or a video call, if you wish.

*****IMPORTANT NOTE:

We perform an initial quality control of all revised manuscripts before re-review. Your manuscript will FAIL this control and the handling will be delayed IN CASE the following APPLIES:

- 1) A data availability section providing access to data deposited in public databases is missing. If you have not deposited any data, please add a sentence to the data availability section that explains that.
- 2) Your manuscript contains statistics and error bars based on $n=2$. Please use scatter blots in these cases. No statistics should be calculated if $n=2$.

When submitting your revised manuscript, please carefully review the instructions that follow below. Failure to include requested items will delay the evaluation of your revision. *****

- 1) a .docx formatted version of the manuscript text (including legends for main figures, EV figures and tables). Please make sure that the changes are highlighted to be clearly visible.
- 2) individual production quality figure files as .eps, .tif, .jpg (one file per figure). Please download our Figure Preparation Guidelines (figure preparation pdf) from our Author Guidelines pages <https://www.embopress.org/page/journal/14693178/authorguide> for more info on how to prepare your figures.
- 3) a .docx formatted letter INCLUDING the reviewers' reports and your detailed point-by-point responses to their comments. As part of the EMBO Press transparent editorial process, the point-by-point response is part of the Review Process File (RPF), which will be published alongside your paper.
- 4) a complete author checklist, which you can download from our author guidelines (<<https://www.embopress.org/page/journal/14693178/authorguide>>). Please insert information in the checklist that is also reflected in the manuscript. The completed author checklist will also be part of the RPF.
- 5) Please note that all corresponding authors are required to supply an ORCID ID for their name upon submission of a revised manuscript (<<https://orcid.org/>>). Please find instructions on how to link your ORCID ID to your account in our manuscript tracking system in our Author guidelines (<<https://www.embopress.org/page/journal/14693178/authorguide#authorshipguidelines>>)
- 6) We replaced Supplementary Information with Expanded View (EV) Figures and Tables that are collapsible/expandable online. A maximum of 5 EV Figures can be typeset. EV Figures should be cited as "Figure EV1, Figure EV2" etc... in the text and their

respective legends should be included in the main text after the legends of regular figures.

7) Please note that a Data Availability section at the end of Materials and Methods is now mandatory. In case you have no data that requires deposition in a public database, please state so instead of refereeing to the database. See also < <https://www.embopress.org/page/journal/14693178/authorguide#dataavailability>>. Please note that the Data Availability Section is restricted to new primary data that are part of this study.

Additional information on source data and instruction on how to label the files are available <<https://www.embopress.org/page/journal/14693178/authorguide#sourcedata>>.

10) Figure legends and data quantification:

- the name of the statistical test used to generate error bars and P values,
 - the number (n) of independent experiments (please specify technical or biological replicates) underlying each data point,
 - the nature of the bars and error bars (s.d., s.e.m.)
- If the data are obtained from n {less than or equal to} 5, show the individual data points in addition to the SD or SEM.
- If the data are obtained from n {less than or equal to} 2, use scatter blots showing the individual data points.

11) Our journal encourages inclusion of *data citations in the reference list* to directly cite datasets that were re-used and obtained from public databases. Data citations in the article text are distinct from normal bibliographical citations and should directly link to the database records from which the data can be accessed. In the main text, data citations are formatted as follows: "Data ref: Smith et al, 2001" or "Data ref: NCBI Sequence Read Archive PRJNA342805, 2017". In the Reference list, data citations must be labeled with "[DATASET]". A data reference must provide the database name, accession number/identifiers and a resolvable link to the landing page from which the data can be accessed at the end of the reference. Further instructions are available at <<https://www.embopress.org/page/journal/14693178/authorguide#referencesformat>>.

12) All Materials and Methods need to be described in the main text using our 'Structured Methods' format. According to this format, the Methods section includes a Reagents and Tools Table (listing key reagents, experimental models, software and relevant equipment and including their sources and relevant identifiers) followed by a Methods and Protocols section describing the methods, ideally using a step-by-step protocol format. The aim is to facilitate adoption of the methodologies across labs. Please download and fill our Reagents and Tools Table template (.docx), which you can find in our author guidelines: <https://www.embopress.org/page/journal/14693178/authorguide#structuredmethods>.

13) As part of the EMBO publication's Transparent Editorial Process, EMBO Reports publishes online a Review Process File to accompany accepted manuscripts. This File will be published in conjunction with your paper and will include the referee reports, your point-by-point response and all pertinent correspondence relating to the manuscript.

Yours sincerely,

=====

Referee #1:

This manuscript provides new insight into the transient permeabilization of the host cell membrane that occurs during *Toxoplasma* invasion. The process is of fundamental interest as it presumably forms the conduit for transfer of ROP proteins into the host cell. Sophisticated electrophysiological assays and data analyses form the basis for the study which is nicely explained and illustrated. I especially appreciated the background section in the Introduction that addresses the topological problem of how parasite proteins in the secretory system gain access to the host cell cytosol. This is well reasoned and extremely clearly stated and it highlights an important underlying cell biological problem that is often overlooked in the microbiology literature. The authors make a compelling argument for why the transient changes in conductance represent multiple small pores rather than a single large one, or a simple membrane break. I am not sufficiently expert in the areas of membrane conductance and pore architecture to critically assess this argument, but I offer a few comments below for consideration. I hope that their explanations will aid the reader as well as further educate me on the topic.

Major findings and importance:

The authors do an excellent job of describing the electrophysical properties of the membrane conductance changes that occur during parasite invasion. They allude to the presence of membranous pores, but in fact these elusive molecules are not defined. The Discussion describes in detail other pore forming proteins in parasites (this was rather lengthy but I enjoyed reading it), none of which are candidates for the current findings. Therefore, I wondered with the advent of proteomes for rhoptries and the ability to model proteins readily using AlphaFold, would it be possible to identify candidate ROP proteins that have potential pore forming structures? This might be too big of an ask for the current paper, but at least a statement of the potential for this approach in future could be added to the discussion.

1) Minor issue in the Introduction where it is stated: "To invade, two types of secretory organelle are employed". This omits reference to dense granules, presumably because they are discharged later after invasion (although they start very early). Since the authors comment on GRA protein involved in protein translocation later in the Discussion, introducing them in the Introduction as a third class of secretory organelles would be helpful.

2) The authors argue based on the electrical properties of quantal changes in conductance that the process of membrane barrier disruption is due to insertion of "proteinaceous pores". I wonder if they can rule out alternative mechanisms such as transient rupture of the membrane followed by resealing? What would the conductance changes look like for disruption of the membrane by insertion of a very fine probe? Since the parasite employs an extendable conoid that engages the host cell during invasion, could this structure create breaks in the membrane that explain the conductance changes? Some discussion of this possibility might aid the reader.

3) The findings nicely present the timing of rapid onset, duration followed by resealing of the "pores" based on analysis of conductance changes. The Discussion speculates about possible repair mechanisms, and in this light it might be interesting to examine whether proteins known to repair membrane break are either recruited to the site of invasion or if loss of these components would alter the pore characteristics and/or invasion. I am thinking of proteins like ESCRT, synaptogamins, annexins

etc. Could the authors speculate on this possibility?

4) The authors nicely define the duration of the open state of the pores as ~ 0.5 sec. This is a relatively short time for the transfer of ROP protein to the host cytosol, which must occur in a single bolus if they in fact transfer only through this open part of the membrane. Can the authors speculate on whether this could occur by diffusion or if it would need to be directed? Alternatively, is it possible some ROP proteins cross intact membranes, perhaps using physical properties similar to bacterial and viral proteins that do so?

Referee #2:

The manuscript by Kegawa et al addresses the long-standing question of how *T. gondii* is able to invade the host cell, causing fast permeation and resealing of the plasma membrane. The authors use high-resolution patch-clamp experiments to show that parasite contact with the host cell causes a fast transient that consistently shows a peak conductance of approximately 3 nS and a duration of around 300 ms. These transients may or may not be followed by tachyzoite invasion, as seen in RON2-KD parasites that are still able to elicit similar transients but fail to form a functional moving junction and invade host cells. Analysis of the quantal conductance indicates a step-wise peak of 0.26 nS that shows no changes under high or low calcium conditions, suggesting that the events are not calcium dependent, and, as indicated by the authors, argues against calcium being the main driver of the conductance.

The work presented here is technically sound and adds some evidence to previous publications describing early changes in membrane conductance, prior to the invasion event, but falls short to identify the nature of the pore. The authors propose that rhoptry secretion is required to form a "pore" (some evidence provided in a companion paper) but at the same time indicate that their analysis suggests this is not a "proteinaceous pore", which is confusing.

In Figure 6, they further explain a model where secretion of rhoptries will be the source of the "pore" but this is speculative and seems to contradict some of their electrophysiology results. To test this hypothesis, a cleaner system could be used, like a planar bilayer recording set up where parasites are added to one side and the formation of the transients can be monitored upon rhoptry secretion. Otherwise, the mysterious nature of the pore remains obscure.

Alternatively, electrophysiological recordings of the mutants used in the companion paper by Male et al, could be useful to clarify the role of secreted proteins in the formation of the pore, providing mechanistic insight on the coupling of the permeation and invasion process and a more solid support to the data presented here.

A hypothesis that is not considered is the possibility that the transient is generated by the mechanical changes of the host lipids caused by the tachyzoites pushing on the host's plasma membrane and prior to the rhoptry secretion. This could cause an increase in hydrophobic effect leading to the initial formation of a pore in absence of protein secretion. Is there any evidence discarding this type of mechanism to generate the initial transient?

Yuto et al

Referee #1:

This manuscript provides new insight into the transient permeabilization of the host cell membrane that occurs during *Toxoplasma* invasion. The process is of fundamental interest as it presumably forms the conduit for transfer of ROP proteins into the host cell. Sophisticated electrophysiological assays and data analyses form the basis for the study which is nicely explained and illustrated. I especially appreciated the background section in the Introduction that possess the topological problem of how parasite proteins in the secretory system gain access to the host cell cytosol. This is well reasoned and extremely clearly stated and it highlights an important underlying cell biological problem that is often overlooked in the microbiology literature. The authors make a compelling argument for why the transient changes in conductance represent multiple small pores rather than a single large one, or a simple membrane break. I am not sufficiently expert in the areas of membrane conductance and pore architecture to critically assess this argument, but I offer a few comments below for consideration. I hope that their explanations will aid the reader as well as further educate me on the topic.

We enthusiastically thank the reviewer for their kind words and encouragement. We appreciate your taking time to critique and give advice towards strengthening this manuscript. We hope that you find this revised manuscript reflects your valuable input into this process.

Major findings and importance:

The authors do an excellent job of describing the electrophysical properties of the membrane conductance changes that occur during parasite invasion. They allude to the presence of membranous pores, but in fact these elusive molecules are not defined. The Discussion describes in detail other pore forming proteins in parasites (this was rather lengthy but I enjoyed reading it), none of which are candidates for the current findings. Therefore, I wondered with the advent of proteomes for rhoptries and the ability to model proteins readily using AlphaFold, would it be possible to identify candidate ROP proteins that have potential pore forming structures? This might be too big of an ask for the current paper, but at least a statement of the potential for this approach in future could be added to the discussion.

Thank you very much for your suggestion. We agree that a bioinformatics analysis including the use of AlphaFold and related tools can help to visualize structures that might be consistent with our measurements and possess the ability to translocate proteins across a lipid bilayer. Therefore we have included new text in the discussion (Lines 386-392) to explain our current approach to identify candidates for future work, and we now reference a recent review that outlines four major protein translocators with very different molecular structures and strategies from studies on the cell biology of protein synthesis, retro-translocation, AAA ATPases,

and peroxisome protein import (Tom A. Rapoport, "A life of Translocations" Annu. Rev. Biochem. 2024. 93:1–20).

To help in identifying proteins responsible for pore formation, bioinformatic analyses including screening for pore-forming structural motifs like amphipathic alpha helices and cell-penetrating peptides, combined with protein structure prediction methodologies such as AlphaFold, are currently underway. In addition to the above pore formation motifs, protein translocon features are also under consideration (Rapoport, 2024).

1) Minor issue in the Introduction where it is stated: "To invade, two types of secretory organelle are employed". This omits reference to dense granules, presumably because they are discharged later after invasion (although they start very early). Since the authors comment on GRA protein involved in protein translocation later in the Discussion, introducing them in the Introduction as a third class of secretory organelles would be helpful.

Thank you for pointing this out. We agree with the importance of dense granules for Apicomplexa and have now included a description in the introduction. We now describe dense granules (DGs) as another secretory organelle (Lines 49-53). While DGs are essential for Toxoplasma biology, the role of DG proteins during the invasion remains unclear. Therefore, we decided to keep the sentence from the original submission (L37-38): "To invade, two types of secretory organelle are employed". However, we revised the introduction to include DGs as a potential source of secreted proteins that may be important in the invasion process (Lines 49-53).

Proteins secreted from a third group of secretory organelles, the dense granules, play an important role in parasite intracellular survival and pathogenesis (Gold et al., 2015, Griffith et al., 2022, Bitew et al., 2024) but there is no evidence that these secreted GRA proteins function in the early stages of invasion.

2) The authors argue based on the electrical properties of quantal changes in conductance that the process of membrane barrier disruption is due to insertion of "proteinaceous pores". I wonder if they can rule out alternative mechanisms such as transient rupture of the membrane followed by resealing?

We thank the reviewer for bringing up this important question. You are correct, we cannot rule out alternative mechanisms such as transient rupture of the membrane followed by resealing. We tried to be clear that the data was consistent with a model, and we were careful not to say it was the only model that the data was consistent with. In the first paragraph of the discussion, we contrast a protein pore with a lipidic pore. As we did not make this point sufficiently clear, we have now revised the text (Lines 230-234).

Ion movement (flux) is likely due to the appearance in the host cell membrane of an aqueous pathway. Pathways for passive ion flux across membranes can be classified as either protein-lined channels (termed "protein pores") or localized ruptures of the hydrocarbon continuity of the lipid bilayer (termed "lipidic pores").

The reason that the alternative model of rupture followed by resealing was not highlighted was because we think our data does not fit as well to this interpretation without making the rupture model more complex. To support quantal changes, transient rupture events would need to be tuned such that each rupture event had similar properties (size and duration). Without scaffolding (which would probably end up being proteinaceous), the diameter of a rupture in a membrane (which we usually term a 'lipidic pore' to contrast it to a 'proteinaceous pore' is a continuous property that would need to be constrained by the balance of membrane lateral and surface tensions (again, assuming no protein involvement). This model is hard to reconcile with the larger conductance steps observed that are most easily explained as multiple, localized unitary events. If these multiple, localized unitary events were ruptures, the tuning required to make them all similar would represent a stabilization that would tend to prevent resealing. Of course, if a new resealing event was triggered by the calcium entry that might be OK, but current models of membrane resealing involve endocytosis of a larger area of membrane that would likely remove the moving junction, and that does not happen.

What would the conductance changes look like for disruption of the membrane by insertion of a very fine probe?

Thank you for your interesting question. Before the patch clamp pipette, electrophysiologists did use fine probes to access the electrical interior of a cell. These were very fine glass capillaries that were drawn to very small diameters. Interestingly, no appreciable conductance was created by the insertion of a pipette other than that of its lumen. From this experimental observation we can conclude that there are no measurable conductance changes associated with the insertion of a very fine glass probe (see Betrag, J. Gen. Physiol. 2017 Vol. 149 No. 4 417–430, <https://doi.org/10.1085/jgp.201611634>)

Since the parasite employs an extendable conoid that engages the host cell during invasion, could this structure create breaks in the membrane that explain the conductance changes?

We thank the reviewer for this important question. Protrusion of the conoid has been discussed as one of the possible forces that might cause membrane breakdown during invasion. We do not think it likely that the conoid is directly

involved in the transient in the conductance presented here because our companion paper (Male et al.,) showed Nd9 and FER2 depleted parasites (normal conoid extension) fail to permeabilize the host cell membrane. However, the conoid extension may alter the membrane tension within the moving junction ring, which could have aided in the bilayer insertion of any putative pore-forming proteins. While we considered this possibility, the lack of a major effect on the rising phase of the transient for the RON2 KD makes this less likely. Thanks to the Reviewer's comment we have added that there is a need to evaluate effects of conoid extension (Lines 410-413):

While outside the scope of this paper, future biophysical study is also needed to evaluate possible effects of conoid extrusion on invasion pore insertion or function; mutants that interrupt the apical polar ring and invasion but retain rhoptry secretion are of interest (Ren et al., 2024).

3) The findings nicely present the timing of rapid onset, duration followed by resealing of the "pores" based on analysis of conductance changes. The Discussion speculates about possible repair mechanisms, and in this light it might be interesting to examine whether proteins known to repair membrane break are either recruited to the site of invasion or if loss of these components would alter the pore characteristics and/or invasion. I am thinking of proteins like ESCRT, synaptogamins, annexins etc. Could the authors speculate on this possibility?

Thank you for providing these insights. We have incorporated your comments at the end of discussion (Lines 403-406):

The recruitment of host cell proteins implicated in membrane repair and resealing, such as ESCRT proteins or dysferlin, may be a useful approach to test the hypothesis that pore closure involves a host cell response.

4) The authors nicely define the duration of the open state of the pores as ~ 0.5 sec. This is a relatively short time for the transfer of ROP protein to the host cytosol, which must occur in a single bolus if they in fact transfer only through this open part of the membrane. Can the authors speculate on whether this could occur by diffusion or if it would need to be directed?

We thank the reviewer for allowing us to comment on the most critical question: Are the pores large enough to allow sufficient ROP protein to transit into the cell via diffusion? We very much wish that we had some experimental data on this topic. Instead, we can provide here the results of a calculation for the best possible speed, if the ROP proteins do not interact with the inner wall of the putative quantal channels. Assuming no interactions, we used one-dimensional flux across the length of the membrane using Einstein's diffusion equation ($x^2 = 2Dt$, where x is the distance moved, D is the diffusion constant, and t is time). Across the membrane is

a short distance to move ($x = 5 \text{ nm} = 5 \times 10^{-3} \mu\text{m}$). This means that even the slowest large protein in cytoplasm having a D (diffusion constant) of $1 \mu\text{m}^2/\text{s}$ would take $25/2 \times 10^{-6} \text{ s} \approx 12 \mu\text{s}$ (0.01 ms). So, there is enough time for 20,000 proteins to move across each putative invasion pore or over 300,000 copies across 10 invasion pores. But if there are interactions with the walls of the pore, competition with macromolecules with reverse gradients (i.e. those that could come out) or if there needs to be time to unravel a globular protein for single file diffusion, it can take longer. It has been suggested to us that the quanta represent fusion of smaller pores into larger ones, to grow in steps, and perhaps accommodate the largest folded ROP proteins that may have to get in (we do not know the folding state of the ROP proteins inside the rhoptry). Having multiple pores is beneficial and compares with multiple lanes on a highway: there is no fighting to get into just one, you can always go to another lane. We call this advantage the “access resistance”. Another issue is that we do not know the diameter of each putative invasion pore because they may be only open when partially occluded by transiting ROP proteins. Given the uncertainties spelled out above, we do not think placing the above analysis in the discussion will advance our understanding enough and are better suited to the coffee or beer break of a conference.

Alternatively, is it possible some ROP proteins cross intact membranes, perhaps using physical properties similar to bacterial and viral proteins that do so?

We thank the reviewer for this interesting question. We did search for some of these motifs but did not find them. For example, one feature can be a long string of arginine found in many CPP (Cell penetrating proteins). Substituting lysine for arginine does not seem to satisfy the molecular condition for transport, so the requirements are not merely charge-specific. This is a very interesting area for future work that we will investigate. In this paper we do not measure protein transfer across the membrane, only ions. We now have added the idea of a CPP motif to the discussion (L389).

Referee #2:

The manuscript by Kegawa et al addresses the long-standing question of how *T. gondii* is able to invade the host cell, causing fast permeation and resealing of the plasma membrane. The authors use high-resolution patch-clamp experiments to show that parasite contact with the host cell causes a fast transient that consistently shows a peak conductance of approximately 3 nS and a duration of around 300 ms. These transients may or may not be followed by tachyzoite invasion, as seen in RON2-KD parasites that are still able to elicit similar transients but fail to form a functional moving junction and invade host cells.

Analysis of the quantal conductance indicate a step-wise peak of 0.26 nS that shows no

changes under high or low calcium conditions, suggesting that the events are not calcium dependent, and, as indicated by the authors, argues against calcium being the main driver of the conductance.

The work presented here is technically sound and adds some evidence to previous publications describing early changes in membrane conductance, prior to the invasion event, but falls short to identify the nature of the pore.

We thank the reviewer. Certainly, we tried to make it clear that we have no evidence for any molecular nature of the pore. Our goal is to describe the job description of the pore, i.e. the physiological phenotype in space and time and the relationship between the pore phenotype and one moving junction component (RON2). We have introduced new methodologies for asking if a small number of quantal pore elements can explain a complicated time-resolved conductance recording.

The authors propose that rhoptry secretion is required to form a "pore" (some evidence provided in a companion paper) but at the same time indicate that their analysis suggest this is not a "proteinaceous pore", which is confusing.

We are indebted to the reviewer for pointing out to us the way our language is confusing to readers. One source of confusion arises from our attempt to focus on single vs. multiple pores. We now separate the definition of a protein pore from a lipidic pore and avoid the more cumbersome term 'proteinaceous'. (Lines 230-234)

Ion movement (flux) is likely due to the appearance in the host cell membrane of an aqueous pathway. Pathways for passive ion flux across membranes can be classified as either protein-lined channels (termed "protein pores") or localized ruptures of the hydrocarbon continuity of the lipid bilayer (termed "lipidic pores").

Another source of confusion is the lack of clarity in discussing the kinetics of single channels and the possibility that there is a single channel that opens with small subconductance steps that continue to flicker at the plateau phase and then step down. We now address this possibility head-on as an alternative explanation. We think that this is less likely because in general subconductance states are not identical in size, but there is always a first time. (see Lines 311-323)

*Alternative explanations for a quantal step size are worth exploring. Multiple steps in conductance are not unusual for proteinaceous pores having subconductance states, such as the voltage-dependent anion channel from *Neurospora* or rat liver (Zimmerberg & Parsegian, 1986). The largest component in the distribution of conductance step sizes for VDAC is the main open-closed transition (also ~ 0.25 nS in salt solutions close to those used here). The fact that the quantal-like conductance changes were the same, irrespective of analytic technique, argues against a single proteinaceous pore*

with 13-16 identical sub-conductance states barring an unusual sequential subconductance state transition pattern that mimics the observed rising and falling time distributions. However, our analysis does not rule out multiple proteinaceous pores with lower numbers of sub-conductance states whose individual stochastic behavior (subconductance states) and combined behavior (activation) results in the macroscopically observed conductance changes.

In Figure 6, they further explain a model where secretion of rhoptries will be the source of the "pore" but this speculative and seem to contradict some of their electrophysiology results.

This is a valid assessment about our proposed model of invasion pores which are induced by rhoptry secretions. We have revised figure 6 to clarify: 1) possible existence of pore-forming agents released from the parasite, and 2) the process of the pore-forming agent self-insertion.

To test this hypothesis, a cleaner system could be used, like a planar bilayer recording set up where parasites are added to one side and the formation of the transients can be monitored upon rhoptry secretion. Otherwise, the mysterious nature of the pore remains obscure.

*Thank you! We would love to do this experiment, and we agree that it would be very informative. Conditions for the invasion of a vesicle by *T. gondii* have been studied but to our knowledge (mostly hearsay) it does not work. Perhaps there are host cell interactions required for invasion; perhaps we lack knowledge of a suitable method (i.e. agonist) to trigger rhoptry secretion without a host cell. We will continue to think about attempting this elegant approach.*

Alternatively, electrophysiological recordings of the mutants used in the companion paper by Male et al, could be useful to clarify the role of secreted proteins in the formation of the pore, providing mechanistic insight on the coupling of the permeation and invasion process and a more solid support to the data presented here.

Thank you for pointing this out. We did many times try to evidence the 'aberrant' calcium spikes discussed in our companion paper (Male and Kegawa et al.) Testing mutants in electrophysiological recording is a powerful method to analyze pore formation events with very high resolution. However, those mutants gave very low numbers of calcium entry transients, so collecting enough transients by the patch clamp technique would mostly fail. In fact, the reason we developed the alternative Ca influx assay was to provide as a high throughput assay (details are presented in our companion paper (Male and Kegawa, et al.).

A hypothesis that is not considered is the possibility that the transient is generated by the mechanical changes of the host lipids caused by the tachyzoites pushing on the host's plasma membrane and prior to the rhoptry secretion. This could cause an increase in hydrophobic defect leading to the initial formation of a pore in absence of protein secretion. Is there any evidence discarding this type of mechanism to generate the initial transient?

We thank the reviewer for this idea! Certainly, pushing on the host cell membrane would increase lipid stress making the formation of a lipid pore more likely. This may also happen upon conoid extension once the moving junction forms. Gliding motility of the parasite on the host cell lipid bilayer is one of the most significant events prior to the rhoptry secretion. The transients were not observed at the moment of parasite attachment nor during gliding. We believe the conoid is not directly involved in the transient in the conductance presented here because our companion paper (Male and Kegawa et al.,) showed Nd9 and FER2 depleted parasites (normal conoid extension) fail to permeabilize host cell membrane. However, the conoid extension may alter the membrane tension within the moving junction ring, which could have aided in the bilayer insertion of any putative pore-forming proteins. While we considered this possibility, the lack of a major effect on the rising phase of the transient for the RON2 KD makes this less likely. To draw attention to this important consideration, the following was added to the end of the discussion (Lines 410-413).

While outside the scope of this paper, future biophysical study is also needed to evaluate possible effects of conoid extrusion on invasion pore insertion or function; mutants that interrupt the apical polar ring and invasion but retain rhoptry secretion are of interest (Ren et al., 2024).

Dear Dr. Zimmerberg

Thank you once more for the submission of your revised manuscript to EMBO reports. I have already forwarded the positive referee reports to you and you find them again copied below.

Browsing through the manuscript myself, I noticed a few editorial things that we need before we can proceed with the official acceptance of your study.

- Please provide up to 5 keywords.
- Please update the citation to Male and Kegawa et al to EMBO reports in press.
- In line 82 you mention the RAPS2 mutant that blocks poration, without explaining what RASP2 is. Without having read Male and Kegawa this statement is unclear.
- Please use capital letters to label figure panels.
- The figures seem to have rather low resolution. Please provide high resolution production quality figure files.
- Reagents and Tools table: Please remove the "Instructions" paragraph. You can also delete all sections that are not required, such as Recombinant DNA etc.
- Please provide the conflict of interest statement in a separate 'Disclosure and competing interests statement' paragraph. For more information see <https://www.embopress.org/page/journal/14693178/authorguide#conflictsofinterest>
- The author names on the manuscript title page should be provided as follows: First Name Last Name (instead of Last Name First Name Initial)
- Regarding the Author Contributions, we now use CRediT to specify the contributions of each author in the journal submission system. Therefore, please remove the Author Contributions from the manuscript file and make sure that the author contributions in our online manuscript tracking system are correct and up-to-date. The information you specified in the system will be automatically retrieved and typeset into the article. You can enter additional information in the free text box provided, if you wish.
- References: et al needs to be used after 10 author names; DOIs should only be used for preprints and datasets that have not been published yet.
- Author checklist: please complete the missing information/responses in line 4-6, and in the section Materials.
- Materials and Methods should be Methods.
- The nomenclature of EV figure legends needs to be Figure EV1, Figure EV2, etc. instead of Supplementary Figure 1a, Supplementary Figure 1b...; each legend should follow the format of the main figures i.e., include all panels in one paragraph and not have separate titles and paragraphs for each panel.
- Please also update the callouts to EV figures accordingly to Figure EV1, etc.
- Please upload the source data as one zip folder per figure. Thank you.
- Our production/data editors have asked you to clarify several points in the figure legends (see below). Please incorporate these changes in the manuscript and return the revised file with tracked changes with your final manuscript submission.

A) Replicates and error bars:

- Please note that information related to n is missing in the legends of figures 4A-D; supplementary figure 5.
- Please note that the measure of center for the error bars needs to be defined in the legend of figure 2A"
- As a standard procedure, we modify title and abstract to make them more accessible to our general readership. Please find my suggestion below my signature, and modify as you see best fit.
- Finally, EMBO Reports papers are accompanied online by
 - A) a short (1-2 sentences) summary of the findings and their significance,
 - B) 2-3 bullet points highlighting key results and

C) a schematic summary figure that provides a sketch of the major findings (not a data image). Please provide the summary figure as a separate file in PNG or JPG format at a size of 550x300-600 pixels (width x height). Please note that the size is rather small and that text needs to be readable at the final size. Please send us this information along with the revised manuscript.

With kind regards,

Martina

=====

Referee #1:

The authors have provide insightful replies and made useful updates to the manuscript. I have no further concerns.

Referee #2:

I would like to thank the authors for carefully and enthusiastically addressing the points of concern raised in the previous review. This manuscript makes a significant contribution to our current understanding of the host cell invasion process by *Toxoplasma gondii*.

=====

Proposed abstract

The obligate intracellular parasite *Toxoplasma gondii* invades its host cell only after secreting proteins required for host cell attachment and entry. The moving junction protein RON2 is inserted into the host membrane and considered important for host cell poration. Here, by acquiring electrophysiological recordings of host cells at sub-200 s resolution, we detect and analyse a transient increase in host membrane conductance following parasite exposure. Transients always precede invasion but parasites depleted of RON2 generate transients without invading, ruling out an essential role for RON2 in generating the conductance pathway. Time-series analysis developed for transients and applied to the entire transient dataset (910,000 data points) reveals multiple quantal conductance changes in the parasite-induced transient, consistent with a rapid insertion, then slower removal, blocking, or inactivation of [potential?] pore components. Quantal steps for wild-type RH strain parasites have a principal mode with Gaussian mean of 0.26 nS, similar in step size to the pore forming protein EXP2, part of the PTEX translocon of malaria parasites. Without RON2 the quantal mean (0.19 nS) is significantly different. Because we observe no parasite invasion without poration, the term "invasion pore" is proposed to describe this transient breach in host cell membrane barrier integrity during invasion.

The authors addressed the remaining editorial issues.

Joshua Zimmerberg
NIH/NICHD
LCMB
Bldg. 10, Room 10D14
10 Center Drive
Bethesda, MD 20892
United States

Dear Dr. Zimmerberg,

I am very pleased to accept your manuscript for publication in the next available issue of EMBO reports. Thank you for your contribution to our journal.

Yours sincerely,
